# LEARNING TO COMPUTE GRÖBNER BASES

## ABSTRACT

Solving a polynomial system, or computing an associated Gröbner basis, has been a fundamental task in computational algebra. However, it is also known for its notoriously expensive computational cost—doubly exponential time complexity in the number of variables in the worst case. In this paper, we achieve for the first time Gröbner basis computation through the training of a transformer. The training requires many pairs of a polynomial system and the associated Gröbner basis, thus motivating us to address two novel algebraic problems: random generation of Gröbner bases and the transformation of them into non-Gröbner polynomial systems, termed as *backward Gröbner problem*. We resolve these problems with zero-dimensional radical ideals, the ideals appearing in various applications. The experiments show that in the five-variate case, the proposed dataset generation method is five orders of magnitude faster than a naive approach, overcoming a crucial challenge in learning to compute Gröbner bases.

## 1 INTRODUCTION

Understanding the properties of polynomial systems and solving them have been a fundamental problem in computational algebra and algebraic geometry with vast applications in cryptography (Bard, 2009; Yasuda et al., 2015), control theory (Park & Regensburger, 2007), statistics (Diaconis & Sturmfels, 1998; Hibi, 2014), computer vision (Stewenius, 2005), systems biology (Laubenbacher & Sturmfels, 2009), and so forth. Special sets of polynomials, called Gröbner bases, play a key role to this end. In linear algebra, the Gaussian elimination simplifies or solves a system of linear equations by transforming its coefficient matrix into the reduced row echelon form. Similarly, a Gröbner basis can be regarded as a reduced form of a given polynomial system, and its computation is a generalization of the Gaussian elimination to general polynomial systems.

However, computing a Gröbner basis is known for its notoriously bad computational cost in theory and practice. It is an NP-hard problem, and its worst-case time complexity is *doubly* exponential in the number of variables (Mayr & Meyer, 1982; Dubé, 1990). Nevertheless, because of its importance, various computation algorithms have been proposed in computational algebra to obtain Gröbner bases in better runtime. Examples include Faugère's F4/F5 algorithms (Faugère, 1999; Faugère, 2002) and M4GB (Makarim & Stevens, 2017).

In this study, we address this challenging task of computing Gröbner bases using machine learning. Recent studies have revealed the potential of transformer models as a powerful solver of algebraic computations. Namely, instead of designing explicit computational procedures, we train a machine learning model using a large amount of (non-Gröbner set, Gröbner basis) pairs. A similar framework has also been used for other mathematics tasks. For example, (Lample & Charton, 2019) showed that transformer models can learn symbolic integration simply by observing many $(\mathrm{d}f/\mathrm{d}x, f)$ pairs in training. The training samples are generated by first randomly generating $f$ and computing its derivative $\mathrm{d}f/\mathrm{d}x$ and/or the reverse process. Other examples include solving ordinary differential equations (Lample & Charton, 2019), symbolic regression (Biggio et al., 2021), and basic linear algebra (Charton, 2022a). For all the tasks, transformers exhibited a high performance.

A crucial challenge in the learning of Gröbner basis computation is that it is mathematically unknown how to efficiently generate many (non-Gröbner set, Gröbner basis) pairs. To resolve this, we need an efficient backward approach (i.e., *solution-to-problem* computation) because, as discussed above, the forward approach (i.e., *problem-to-solution* computation) is prohibitively expensive. Particularly, we need to address (i) a random generation of Gröbner bases and (ii) a backward transfor-

mation from a Gröbner basis to an associated non-Gröbner set. To our knowledge, neither of them has been addressed in computational algebra because of the lack of motivations and applications; all the efforts have been dedicated to the forward computation from a non-Gröbner set to Gröbner basis. The central contributions of this study are the discovery of these new mathematical problems, efficient computational algorithms for them, and the first machine learning approach for Gröbner basis computation, summarized as follows.

- We provide the first approach to the Gröbner computation using a machine learning model and experimentally show that transformers can actually learn it. Unlike most prior studies, our results indicate that training a transformer may be a compromise to NP-hard problems to which no efficient (even approximate or probabilistic) algorithms have been designed.

- We uncovered two unexplored algebraic problems—random generation of Gröbner bases and backward Gröbner problem and propose efficient methods to address them, particularly for the zero-dimensional case. These problems are not only essential to generating a dataset for training a transformer but also algebraically interesting, thus encouraging the interaction between computational algebra and machine learning.

- Our experiments demonstrate that our method is highly efficient and faster than a baseline method by two to five orders of magnitude in dataset generation. Transformers trained on the generated datasets successfully compute Gröbner bases with moderate accuracy.

## 2 RELATED WORK

**Gröbner basis computation.** Gröbner basis is one of the fundamental concepts in algebraic geometry and commutative ring theory, see (Cox, 2005; Gruel & Pfister, 2008). By its computational aspect, Gröbner basis is a very useful tool for analyzing the mathematical structures of solutions of algebraic constraints. Notably, the form of Gröbner bases is very suited for finding solutions and allows parametric coefficients. For such sought-after applications, it is vital to make Gröbner basis computation very efficient and practical. Following the definition of Gröbner bases in (Buchberger, 1976), the original algorithm to compute them can be presented as (i) create potential new leading terms by constructing *S-polynomials*, (ii) reduce them either to zero or to new polynomials for the Gröbner basis, and (iii) repeat until no new S-polynomials can be constructed. Plenty of works has been developed to surpass this algorithm. There are four main strategies: (a) avoid unnecessary S-polynomials, following the $F_5$ algorithm and the more general *signature-based algorithms* (Faugère, 2002; Bardet et al., 2015). Machine learning appeared for this task in (Peifer et al., 2020). (b) more efficient reduction using efficient linear algebraic computations, following (Faugère, 1999) and the very recent GPU-using (Berthomieu et al., 2023). (c) perform *modular computations*, following (Arnold, 2003; Noro & Yokoyama, 2018), to prevent *coefficient growth* during the computation. (d) use the structure of the ideal, e.g., (Faugère et al., 1993; Berthomieu et al., 2022) for change of term ordering for zero-dimensional ideals or (Traverso, 1997) when the Hilbert function is known. In this study, we presents the fifth strategy: (e) Gröbner computation fully via learning without specifying any mathematical procedures.

**Math transformers.** Recent studies have revealed that transformers can be used for mathematical reasoning and symbolic computation. The training of such *math transformers* only requires samples (i.e., problem–solution pairs), and no explicit procedures need to be specified. (Lample & Charton, 2019) presented the first study that uses transformers for mathematical problems. They showed that transformers can learn symbolic integration and differential equation solving with training with sufficiently many and diverse samples. Since then, transformers have been applied to checking local stability and controllability of differential equations (Charton et al., 2021), polynomial simplification (Agarwal et al., 2021), linear algebra (Charton, 2022a;b), attack to the LWE cryptography Wenger et al. (2022); Li et al. (2023), and symbolic regression (Biggio et al., 2021; Kamienny et al., 2022; d'Ascoli et al., 2022; Kamienny et al., 2023). (Saxton et al., 2019) provides comprehensive experiments over various mathematical tasks. Another line of math transformers is for automatic theorem proving (Liu et al., 2022; Palermo et al., 2022; Wang & Deng, 2020; Lample et al., 2022; Li et al., 2021; Polu & Sutskever, 2020; Li et al., 2020; Saxton et al., 2019), which is not of our current scope. Several recent studies point out that transformers may only perform well for in-distribution samples and not generalize well to out-distributions (Dziri et al., 2023). There

are several transformer models that address this challenge (Kim et al., 2021; Cognolato & Testolin, 2022). While this is a fundamental challenge of the math transformers, the focus of our study lies in revealing whether transformers can learn an extremely difficult mathematical problem, Gröbner basis computation, and establish a dataset generation method to experimentally examine it.

## 3    NOTATIONS AND DEFINITIONS

Throughout the paper, we consider a polynomial ring $k[x_1, \ldots, x_n]$ with a field $k$ and variables $x_1, \ldots, x_n$.[1] We here introduce basic definitions in algebra. Refer to App. A as necessary.

**Definition 3.1** (Affine variety). Let $F = \{f_1, \ldots, f_s\}$ be a subset of $k[x_1, \ldots, x_n]$. Then, we set

$$V(F) = V(f_1, \ldots, f_s) = \{p \in k^n \mid f_1(p) = \cdots = f_s(p) = 0\}. \tag{3.1}$$

We call this the *affine variety* defined by $f_1, \ldots, f_s$ (or by $F$).

Namely, the affine variety $V(f_1, \ldots, f_s)$ is the zero set of the polynomial system $f_1(x_1, \ldots, x_n) = \cdots = f_s(x_1, \ldots, x_n) = 0$. Next, we introduce a special set of polynomials.

**Definition 3.2** (Ideal). A subset $I \subset k[x_1, \ldots, x_n]$ is an *ideal* if it satisfies the following: (i) $0 \in I$, (ii) if $f, g \in I$, then $f + g \in I$, and (iii) if $f \in I$ and $h \in k[x_1, \ldots, x_n]$, then $hf \in I$.

An ideal $I$ that contains $f_1, \ldots, f_s$ relates to the polynomial system $f_1(x_1, \ldots, x_n) = \cdots = f_s(x_1, \ldots, x_n) = 0$. Indeed, for any $p \in k^n$, $f_i(p) = f_j(p) = 0$ implies $(f_i + f_j)(p) = 0$, and for any $h \in k[x_1, \ldots, x_n]$, $(hf_i)(p) = 0$ holds. Thus, roughly speaking, $I$ contains all the polynomials that can be freely appended to the system $f_1(p) = \cdots = f_s(p) = 0$ without changing its affine variety (i.e., solutions). This idea is formalized as follows.

**Definition 3.3** (Generators). For $F = \{f_1, \ldots, f_s\} \subset k[x_1, \ldots, x_n]$, the following set

$$\langle F \rangle = \langle f_1, \ldots, f_s \rangle = \left\{ \sum_{i=1}^{s} h_i f_i \mid h_1, \ldots, h_s \in k[x_1, \ldots, x_n] \right\}. \tag{3.2}$$

is an ideal and said to be *generated* by $f_1, \ldots, f_s$ (or by $F$), and $f_1, \ldots, f_s$ are called *generators*.

We next define several notions that are necessary to define the Gröbner basis.

**Definition 3.4** (Term order (informal)). A *term order* $\prec$ is an order of terms[2] such as $1 \prec x_n \prec x_{n-1} \prec \cdots$. Constant term $1 = x_1^0 \cdots x_n^0$ is the least prioritized in any term order.

Strictly speaking, a term order satisfies a few more conditions, but we omit the detail as it is unnecessary to follow this study. We provide two well-known term orders to give an intuition.

**Example 3.5.** The *lexicographic order* $\prec_{\text{lex}}$ prioritizes terms with larger exponents for the variables of small indices, e.g.,

$$1 \prec_{\text{lex}} x_n \quad \text{and} \quad x_2 \succ_{\text{lex}} x_3^2 \quad \text{and} \quad x_1 x_2 x_3^2 \prec_{\text{lex}} x_1 x_2^2 x_3. \tag{3.3}$$

Two terms are first compared in terms of the exponent in $x_1$ (larger one is prioritized), and if a tie-break is needed, the next variable $x_2$ is considered, and so forth.

The next notion plays an important role in the Gröbner basis theory.

**Definition 3.6** (Leading term). Let $F = \{f_1, \ldots, f_s\} \subset k[x_1, \ldots, x_n]$ and let $\prec$ be a term order. The leading term $\text{LT}(f_i)$ of $f_i$ is the largest term in $f_i$ in ordering $\prec$. The leading term set of $F$ is $\text{LT}(F) = \{\text{LT}(f_1), \ldots, \text{LT}(f_s)\}$.

Given a polynomial set $F$, the computation of a Gröbner basis is essentially a *simplification* of $F$ without changing the affine variety $V(F)$. For example, when $F = \{h_1 f, h_2 f\}$ with $h_1 + h_2 = 1$, we may *simplify* it to $\tilde{F} = \{f\}$ through polynomial divisions, while maintaining $V(F) = V(\tilde{F})$. The leading term defines the simplicity of polynomials; given two polynomials $f_1, f_2 \in I$, $f_1$ is considered simpler if $\text{LT}(f_1) \prec \text{LT}(f_2)$. Indeed, $\text{LT}(f) \prec \text{LT}(h_1 f), \text{LT}(h_2 f)$ in the above example. Now, we introduce Gröbner bases.

---

[1]One may regard $k[x_1, \ldots, x_n]$ as the set of all $n$-variate polynomials with coefficients in $k$ (e.g., $k = \mathbb{R}$).

[2]*Terms* refer to power products of variables, i.e, $x_1^{\alpha_1} \cdots x_n^{\alpha_n}$ with $\alpha_1, \ldots, \alpha_n \in \mathbb{Z}_{\geq 0}$.

**Definition 3.7** (Gröbner basis). Fix a term order $\prec$. A finite subset $G$ of an ideal $I$ is said to be a $\prec$-*Gröbner* basis of $I$ if $\langle \mathrm{LT}(G) \rangle = \langle \mathrm{LT}(I) \rangle$.

Note that $\langle \mathrm{LT}(G) \rangle \subset \langle \mathrm{LT}(I) \rangle$ is trivial from $G \subset I$. The nontriviality of the Gröbner basis lies in $\langle \mathrm{LT}(G) \rangle \supset \langle \mathrm{LT}(I) \rangle$; that is, a finite number of leading terms can generate the leading term of any polynomial in the infinite set $I$. The Hilbert basis theorem (Cox et al., 2015) guarantees that every ideal $I \neq \{0\}$ has a Gröbner basis.

**Other notations.** The subset $k[x_1, \ldots, x_n]_{\leq d} \subset k[x_1, \ldots, x_n]$ denotes the set of all polynomials of total degree at most $d$. For a polynomial matrix $A \in k[x_1, \ldots, x_n]^{s \times s}$, its determinant is given by $\det(A) \in k[x_1, \ldots, x_n]$. The set $\mathbb{F}_p$ with a prime number $p$ denotes the finite field of order $p$.

We conclude this section with an intuitive (but not necessarily accurate) explanation of Gröbner bases for those unfamiliar with this fundamental concept in algebra.

**Intuitive explanation of Gröbner bases and polynomial system solving.** Let $G = \{g_1, \ldots, g_t\}$ be a Gröbner basis of an ideal $\langle F \rangle = \langle f_1, \ldots, f_s \rangle$. The polynomial system $g_1(x_1, \ldots, x_n) = \cdots = g_t(x_1, \ldots, x_n) = 0$ is a simplified form of $f_1(x_1, \ldots, x_n) = \cdots = f_s(x_1, \ldots, x_n) = 0$ with the same solution set. With a term order $\prec_{\mathrm{lex}}$, $G$ has a form $g_1 \in k[x_{n_1}, \ldots, x_n], g_2 \in k[x_{n_2}, \ldots, x_n], \ldots, g_t \in k[x_{n_t}, \ldots, x_n]$ with $n_1 \leq n_2 \leq \ldots \leq n_t$, which may be regarded as the "reduced row echelon form" of a polynomial system. In our particular case (i.e., zero-dimensional ideals in shape position; cf. Sec. 4.1), we have $(n_1, n_2, \ldots, n_t) = (1, 2, \ldots, n)$. Thus, one can obtain the solutions of the polynomial system using a backward substitution, i.e., by first solving a univariate polynomial $g_t$, next solving bivariate polynomial $g_{t-1}$, which becomes univariate by substituting the solutions of $g_t$, and so forth.

## 4 RANDOM GRÖBNER BASES AND BACKWARD GRÖBNER PROB.

Our goal is to realize Gröbner basis computation through a machine learning model. To train such a model, we need a large training set $\{(F_i, G_i)\}_{i=1}^m$ with a finite polynomial set $F_i \subset k[x_1, \ldots, x_n]$ and a Gröbner basis $G_i$ of $\langle F \rangle$. As the computation of $F_i$ to $G_i$ is computationally expensive in general, we first generate a Gröbner basis $G_i$ randomly and then transform it to non-Gröbner set $F_i$. Such a *backward generation* (i.e., solution-to-problem process) has also been used in related studies that train transformer models for mathematical computations. This is because in general, the forward generation (i.e., problem-to-solution process) is much harder.

What makes the learning of Gröbner basis computation hard is that, to our knowledge, neither (i) a random generation of Gröbner basis nor (ii) the backward transform from Gröbner basis to non-Gröbner set has been considered in computational algebra. A central question in computational algebra has been posed on an efficient Gröbner basis computation (i.e., forward generation), and nothing motivates the random generation of Gröbner basis nor the backward transform. Interestingly, machine learning now sheds light on these problems. Formally, we address the following problems for dataset generation.

**Problem 4.1** (Random generation of Gröbner bases). *Fix a term order. Find a collection $\mathcal{G} = \{G_i\}_{i=1}^m$ of Gröbner bases, where $G_i \subset k[x_1, \ldots, x_n]$ is a Gröbner basis of $\langle G_i \rangle$. The collection should contain diverse bases, and we need an efficient algorithm for constructing them.*

**Problem 4.2** (Backward Gröbner problem). *Fix a term order. Given a Gröbner basis $G \subset k[x_1, \ldots, x_n]$, find a collection $\mathcal{F} = \{F_i\}_{i=1}^\mu$ of polynomial sets that are not Gröbner bases but $\langle F_i \rangle = \langle G \rangle$ for all $i = 1, \ldots, \mu$. The collection should contain diverse sets, and we need an efficient algorithm for constructing them.*

In this paper, we address these problems for zero-dimensional ideals.

**Definition 4.3** (Zero-dimensional ideal). Let $F$ be a set of polynomials in $k[x_1, \ldots, x_n]$. An ideal $\langle F \rangle$ is called a *zero-dimensional ideal* if all but a finite number of terms belong to $\mathrm{LT}(\langle F \rangle)$.

If $k$ is an algebraically closed field (e.g., $k = \mathbb{C}$), zero-dimensional ideals relate to polynomial systems with a finite number of solutions (i.e., $V(F)$ is a finite set.). Zero-dimensional ideals appear in various application of Gröbner bases. For example, a multivariate public-key encrypted communication (a candidate of the Post-Quantum Cryptography) with a public polynomial system $F$ over

a finite field $\mathbb{F}_p$ will be broken if one finds an element of $V(F \cup \{x_1^p - x_1, \ldots, x_n^p - x_n\})$ and the ideal $\langle F \cup \{x_1^p - x_1, \ldots, x_n^p - x_n\} \rangle$ is a zero-dimensional ideal (Ullah, 2012, Sec. 2.2).

A more general approach to Probs. 4.1 and 4.2 with positive-dimensional ideals is an open problem for algebraists. The proofs of the results in the following sections can be found in App. C.

### 4.1 RANDOM GENERATION OF GRÖBNER BASES

To address Prob. 4.1, we propose to use the properties of ideals in *shape position*, for which Gröbner bases have a simple structure. Another potential approach is discussed in App. B.

**Definition 4.4** (Shape position). Ideal $I \subset k[x_1, \ldots, x_n]$ is called in *shape position* if its Gröbner basis with respect to $\prec_{\text{lex}}$ has a form of

$$G = \{h(x_n), x_1 - g_1(x_n), \ldots, x_{n-1} - g_{n-1}(x_n)\}, \tag{4.1}$$

where $h, g_1, \ldots, g_{n-1}$ are univariate polynomials in $k[x_n]$.

As can be seen, the $\prec_{\text{lex}}$-Gröbner basis consists of a univariate polynomial in $x_n$ and the difference of univariate polynomials in $x_n$ and a leading term $x_i$ $(i < n)$. Unfortunately, not all ideals are in shape position. Nevertheless, zero-dimensional radical ideals are almost always in shape position (but the converse is not always true).

**Proposition 4.5** (Gianni & Mora (1989), Prop. 1.6, Noro & Yokoyama (1999), Lem. 4.4). *Let I be a zero-dimensional radical ideal[3]. If k is of characteristic 0 or a finite field of large enough order, then a random linear coordinate change puts I in shape position.*

Thus, we can efficiently generate a random zero-dimensional ideal by sampling $n$ polynomials in $k[x_n]$, i.e., $h, g_1, \ldots, g_{n-1}$ with $h \neq 0$. This resolves Prob. 4.1 even with large $m$.

**Gröbner bases for general term orders.** This approach assumes term order $\prec_{\text{lex}}$. In most practical scenarios, $\prec_{\text{lex}}$-Gröbner bases are already sufficient as they provide a simplified form of polynomial systems as presented at the end of Sec. 3. In computational algebra, it is more common to use the graded reverse lexicographic order because it typically leads to faster computation and smaller size of Gröbner bases. The obtained Gröbner basis is then transformed to the $\prec_{\text{lex}}$-Gröbner basis using an efficient change-of-ordering algorithm such as the FGLM algorithm (Faugère et al., 1993). In our case, if necessary, one can similarly use it to obtain Gröbner bases in non-$\prec_{\text{lex}}$ order from $\prec_{\text{lex}}$ Gröbner bases. The cost of FGLM algorithm (Faugère et al., 1993) is $\mathcal{O}(n \cdot \deg(h)^3)$.[4]

### 4.2 BACKWARD GRÖBNER PROBLEM

To address Prob. 4.2, we consider the following problem.

**Problem 4.6.** *Let $I \subset k[x_1, \ldots, x_n]$ be a zero-dimensional ideal, and let $G = (g_1, \ldots, g_t)^\top \in k[x_1, \ldots, x_n]^t$ be a $\prec$-Gröbner basis of I with respect to term order $\prec$.[5] Find a polynomial matrix $A \in k[x_1, \ldots, x_n]^{s \times t}$ that gives a non-Gröbner set $F = (f_1, \ldots, f_s)^\top = AG$ such that $\langle F \rangle = \langle G \rangle$.*

Namely, we generate a set of polynomials $F = (f_1, \ldots, f_s)^\top$ from $G = (g_1, \ldots, g_t)^\top$ by $f_i = \sum_{j=1}^t a_{ij} g_j$, for $i = 1, \ldots, s$, where $a_{ij} \in k[x_1, \ldots, x_n]$ denotes the $(i, j)$-th entry of $A$. However, $\langle F \rangle$ and $\langle G \rangle$ are generally not identical. The following provides how we should design $A$ to achieve $\langle F \rangle = \langle G \rangle$ for the zero-dimensional case (without radicality or shape position assumption).

**Theorem 4.7.** *Let $G = (g_1, \ldots, g_t)^\top$ be a Gröbner basis of a zero-dimensional ideal in $k[x_1, \ldots, x_n]$. Let $F = (f_1, \ldots, f_s)^\top = AG$ with $A \in k[x_1, \ldots, x_n]^{s \times t}$.*

1. *If $\langle F \rangle = \langle G \rangle$, it implies $s \geq n$.*

2. *If A has a left-inverse in $k[x_1, \ldots, x_n]^{s \times t}$, $\langle F \rangle = \langle G \rangle$ holds.*

3. *The equality $\langle F \rangle = \langle G \rangle$ holds if and only if there exists a matrix $B \in k[x_1, \ldots, x_n]^{t \times s}$ such that each row of $BA - E_t$ is a syzygy[6] of G, where $E_t$ is the identity matrix of size t.*

---

[3]Refer App. A for the definition.

[4]Strictly speaking, the time complexity is here based on the number of arithmetic operations over $k$.

[5]We surcharge notations to mean that the set $\{g_1, \ldots, g_t\}$ defined by the vector $G$ is a $\prec$-Gröbner basis.

[6]Refer to App. A for the definition.

In Thm. 4.7, the first statement argues that polynomial matrix $A$ should have at least $n$ rows to have $\langle F \rangle = \langle G \rangle$. If the ideal is in shape position, we have a $\prec_{\text{lex}}$-Gröbner basis $G$ of size $n$, and thus, $A$ becomes a square or tall matrix. The second statement shows a sufficient condition for designing $A$ such that $\langle AG \rangle = \langle G \rangle$. The third statement provides the sufficient and necessary conditions. In practice, the second statement provides a simple approach for the random transformation of a Gröbner basis to a non-Gröbner set without changing the ideal.

We first assume to use $\prec_{\text{lex}}$, where $G$ has exactly $n$ generators. Discussion on the case with general term orders will come later. For the case of $s = n$, we have the following.

**Proposition 4.8.** *For any $A \in k[x_1, \ldots, x_n]^{n \times n}$ with $\det(A) \in k \setminus \{0\}$, we have $\langle F \rangle = \langle G \rangle$.*

As (non-zero) constant scaling does not change the ideal, we focus on $A$ with $\det(A) = \pm 1$ without loss of generality. Then, such $A$ can be constructed using a Bruhat decomposition in the form of

$$A = U_1 P U_2, \tag{4.2}$$

where $U_1, U_2 \in k[x_1, \ldots, x_n]^{n \times n}$ are upper-triangular matrices with all-one diagonal entries (i.e., unimodular upper-triangular matrices) and $P \in \{0, 1\}^{n \times n}$ denotes a permutation matrix. Noting that $A^{-1}$ satisfies $A^{-1} A = E_n$, we have $\langle AG \rangle = \langle G \rangle$ from Thm. 4.7. Therefore, random sampling $(U_1, U_2, P)$ of unimodular upper-triangular matrices $U_1, U_2$ and a permutation matrix $P$ resolves the backward Gröbner problem for $s = n$.

We extend this idea to the case of $s > n$ using a rectangular unimodular upper-triangular matrix:

$$U_2 = \begin{pmatrix} U_2' \\ O_{s-n,n} \end{pmatrix} \in k[x_1, \ldots, x_n]^{s \times n}, \tag{4.3}$$

where $U_2' \in k[x_1, \ldots, x_n]^{n \times n}$ denotes a unimodular upper-triangular matrix and $O_{s-n,n} \in k[x_1, \ldots, x_n]^{(s-n) \times n}$ is the zero matrix. The permutation matrix is now $P \in \{0, 1\}^{s \times s}$.

Note that $U_2 G$ already gives a non-Gröbner set such that $\langle U_2 G \rangle = \langle G \rangle$; however, the polynomials in the last $s - n$ entries of $U_2 G$ are all zero by its construction. Thus, we need a permutation matrix $P$ to shuffle the rows and also $U_1$ to exclude the zero polynomial from the final polynomial set.

To summarize, our strategy is to compute $F = U_1 P U_2 G$, which only requires a sampling of $\mathcal{O}(s^2)$ polynomials in $k[x_1, \ldots, x_n]$, and $\mathcal{O}(n^2 + s^2)$-times multiplications of polynomials. Note that even in the large polynomial systems given in the MQ challenge, a post-quantum cryptography challenge, we have $n < 100$ and $s < 200$ (Yasuda et al., 2015).

### 4.3 Dataset generation algorithm

Summarizing the discussion in the previous sections, we have Alg. 1 and the following.

**Theorem 4.9.** *Consider polynomial ring $k[x_1, \ldots, x_n]$. Given dataset size $m$, maximum degrees $d, d'$, maximum size of non-Gröbner set $s_{\max} \geq n$, and term order $\prec$, Alg. 1 returns a collection $\mathcal{D} = \{(F_i, G_i)\}_{i=1}^m$ with the following properties: For all $i = 1, \ldots, m$,*

    *1. Both $F_i, G_i \subset k[x_1, \ldots, x_n]$ are finite sets and $|F_i| \leq s_{\max}$.*

    *2. The set $G_i$ is a $\prec$-Gröbner basis. The set $F_i$ is not, unless $G_i, U_1, U_2', P$ are sampled in a non-trivial Zariski closed subset.[7]*

    *3. The ideal $\langle F_i \rangle = \langle G_i \rangle$ is a zero-dimensional ideal in shape position.*

*The time complexity is $\mathcal{O}(m(n S_{1,d} + s^2 S_{n,d'} + (n^2 + s^2) M_{n,2d'+d}))$ when $\prec = \prec_{\text{lex}}$, where $S_{n,d}$ denotes the complexity of sampling an $n$-variate polynomial with total degree at most $d$, and $M_{n,d}$ denotes that of multiplying two $n$-variate polynomials with total degree at most $d$. If $\prec \neq \prec_{\text{lex}}$, $\mathcal{O}(nd^3)$ is additionally needed.*

The proposed dataset generation method is a backward approach, which first generates solutions and then transforms them into problems. In this case, we have control over the complexity of the Gröbner

---

[7]This can happen with probability zero if $k$ is infinite and very low probability over large finite field.

---

**Algorithm 1:** Dataset generation for learning to compute zero-dimensional Gröbner bases.

**Assumption:** polynomial ring $k[x_1, \ldots, x_n]$
**Input:** dataset size $m$, maximum degrees $d, d'$, maximum size of non-Gröbner set $s_{\max} \geq n$,
       and term order $\prec$.
**Output:** collection $\mathcal{D} = \{(F_i, G_i)\}_{i=1}^m$ of non-Gröbner set $F_i \in k[x_1, \ldots, x_n]^m$ and a
       $\prec$-Gröbner basis $G_i \subset k[x_1, \ldots, x_n]$ with $\langle F \rangle = \langle G \rangle$, a zero-dimensional ideal.

---

1   $\mathcal{D} \leftarrow \{\ \}$
2   **for** $i = 1, \ldots, m$ **do**
3      $G_i \leftarrow \{h\}$ with a non-constant polynomial $h$ sampled from $k[x_n]_{\leq d}$.     ▷ `Problem 4.1`
4      **for** $j = 1, \ldots, n - 1$ **do**
5        $G_i \leftarrow G_i \cup \{g_j\}$ with $g_j$ sampled from $k[x_n]_{\leq d}$.
6      **end**
7      $s \sim \mathbb{U}[n, s_{\max}]$                                               ▷ `Problem 4.2`
8      Sample a unimodular upper-triangular matrix $U_1 \in k[x_1, \ldots, x_n]_{\leq d'}^{s \times s}$.
9      Sample a unimodular upper-triangular matrix $U_2' \in k[x_1, \ldots, x_n]_{\leq d'}^{n \times n}$.
10     Sample a permutation matrix $P \in \{0, 1\}^{s \times s}$.
11     $F_i \leftarrow U_1 P U_2 G_i$, where $U_2 = [U_2'^\top \ \ O_{n, s-n}]^\top \in k[x_1, \ldots, x_n]^{s \times n}$.
12     **if** $\prec \ \neq \prec_{\text{lex}}$ **then**
13       $G_i \leftarrow \text{FGLM}(G_i, \prec_{\text{lex}}, \prec)$
14     **end**
15     $\mathcal{D} \leftarrow \mathcal{D} \cup \{(F_i, G_i)\}$              ▷ `Reorder terms in` $F_i$ `if` $\prec \neq \prec_{\text{lex}}$`.`
16   **end**

---

bases and can add some intrinsic structure if any prior information is available. For example, polynomial systems used in a multivariate encryption scheme tend to have only a single solution, and it belongs to the base field (Yasuda et al., 2015) (and thus, are zero-dimensional systems). Ideals generated by such polynomial systems are in shape position and have Gröbner bases of the following form: $G = \langle x_n - a_n, x_1 - a_1, \ldots, x_{n-1} - a_{n-1} \rangle$, where $a_1, a_2, \ldots, a_n$ are constants (Ullah, 2012). Backward approaches allow one to restrict the Gröbner bases in a dataset into such a class.

This is not the case with forward approaches. Instead, they can include prior information into non-Gröbner sets, although it is computationally expensive to obtain the corresponding Gröbner bases from them. It is also worth noting that a naive forward approach, which first randomly generates non-Gröbner sets and then computes their Gröbner bases, should be avoided even when the computational cost could be resolved. In many such cases, a Gröbner basis $G$ computed directly from a random non-Gröbner set $F$ will share some polynomials with $F$ while such a behavior is unusual in various cases where one needs to compute Gröbner bases.

## 5   EXPERIMENTS

We now present experimental results on training a transformer for Gröbner basis computation. All the experiments were conducted on a workstation with 16-core CPUs, 512 GB RAM, and a single NVIDIA RTX A6000 GPU. We provide more information on the profile of generated datasets, the training setup, and additional experimental results in App. D. The code will be available soon.

### 5.1   DATASET GENERATION

We first demonstrate the efficiency of the proposed dataset generation framework. We constructed eight datasets $\{\mathcal{D}_n(\mathbb{F}_p)\}_{n,p}$ for $n \in \{2, 3, 4, 5\}$ and $p \in \{7, 31\}$ and measured the runtime of the forward generation and our backward generation.

**Generation setup of $\mathcal{D}_n(\mathbb{F}_p)$.**   The dataset $\mathcal{D}_n(\mathbb{F}_p)$ consists of 1,000 pairs of non-Gröbner set and Gröbner basis in $\mathbb{F}_p[x_1, \ldots, x_n]$. Each sample $(F, G) \in \mathcal{D}_n(\mathbb{F}_p)$ was prepared using the process given in Alg. 1 with $(d, d', s_{\max}, \prec) = (5, 3, n + 2, \prec_{\text{lex}})$. The number of terms of univariate polynomials and $n$-variate polynomials is uniformly determined from $[1, 5]$ and $[1, 2]$, respectively.

Table 1: Runtime comparison (in seconds/milliseconds) of forward generation (F.) and backward generation (B.) of dataset $\mathcal{D}_n(\mathbb{F}_p)$ of size 1000. The forward generation used either of the three algorithms provided in Sagemath with the libSingular backend. For $n = 4, 5$, we set a timeout limit to one second (added to the total runtime at every occurrence) for each Gröbner basis computation. The numbers with $*$ and $\dagger$ include the timeout for more than 10 % and 40 % of the runs, respectively.

| Method | $\mathbb{F}_7[x_1, \ldots, x_n]$ | | | | $\mathbb{F}_{31}[x_1, \ldots, x_n]$ | | | |
|---|---|---|---|---|---|---|---|---|
| | $n = 2$ | $n = 3$ | $n = 4$ | $n = 5$ | $n = 2$ | $n = 3$ | $n = 4$ | $n = 5$ |
| F. (STD) [sec] | 0.38 | 3.46 | 185* | 430$^\dagger$ | 0.36 | 2.61 | 207* | 437$^\dagger$ |
| F. (SLIMGB) | 0.41 | 5.14 | 174* | 444$^\dagger$ | 0.40 | 7.28 | 202* | 460$^\dagger$ |
| F. (STDFGLM) | 0.97 | 1.36 | 15.5 | 114 | 1.00 | 1.45 | 18.7 | 166 |
| B. (ours) [msec] | **2.05** | **2.88** | **3.70** | **4.90** | **2.11** | **2.78** | **3.47** | **5.27** |

Table 2: Accuracy [%] of Gröbner basis computation by transformers on $\mathcal{D}_n(\mathbb{F}_7)$ with several batch size $B$ and term order. In the support accuracy, two polynomials are considered identical if they consist of an identical set of terms (i.e., identical *support*). The last two column shows the results of training with $\mathcal{D}_2(\mathbb{F}_7)$ using $\prec_{\mathrm{lex}}$-Gröbner bases or $\prec_{\mathrm{grvlex}}$-Gröbner bases. Note that the datasets for $n = 3, 4, 5$ are here constructed using $U_1, U_2'$ (cf. Alg. 1) with density $\sigma = 0.6, 0.3, 0.2$, respectively. Refer to the main text for the details.

| Method | $B = 8$ | | | | $B = 16, n = 2$ | |
|---|---|---|---|---|---|---|
| | $n = 2$ | $n = 3$ | $n = 4$ | $n = 5$ | $\prec_{\mathrm{lex}}$ | $\prec_{\mathrm{grvlex}}$ |
| accuracy | 41.8 | 64.0 | 74.1 | 79.7 | 50.2 | 5.6 |
| support acc. | 72.9 | 84.0 | 88.1 | 87.8 | 74.9 | 13.4 |

**Forward generation.** In the forward generation, one may first generate random polynomial sets and then compute their Gröbner bases. However, this leads to a dataset with a totally different complexity from that constructed by the backward generation, leading to an unfair runtime comparison between the two generation processes. As such, the forward generation instead computes Gröbner bases of the non-Gröbner sets given by the backward generation, leading to the identical dataset. We used SageMath (The Sage Developers, 2023) with the libSingular backend.

In Tab. 1, our backward generation is significant orders of magnitude faster than the forward generation. A sharp runtime growth is observed in the forward generation as the number of variables increases. Note that these numbers only show the runtime on 1,000 samples, while our training requires a million samples. Therefore, the forward generation is almost infeasible, indicating that the proposed method resolves a critical challenge in the learning of Gröbner basis computation.

## 5.2 GRÖBNER BASIS COMPUTATION WITH TRANSFORMERS

We now demonstrate that transformers can learn to compute Gröbner bases. To examine the general transformer's ability, we focus on a standard architecture (e.g., 6 encoder/decoder layers and 8 attention heads) and a standard training setup (e.g., the AdamW optimizer (Loshchilov & Hutter, 2019) with $(\beta_1, \beta_2) = (0.9, 0.999)$ and the linear decay of learning rate from $10^{-4}$). The batch size was set to 8, and models were trained for 10 epochs. Refer to App. D for complete information.

Each polynomial set in the datasets is converted into a sequence in the prefix representation. Unlike natural language processing, our task does not allow the truncation of an input sequence to effectively shorten the input sequences because the first term of the first polynomial in $F$ certainly relates to the last term of the last polynomial. To keep the input sequence length manageable for vanilla transformers, we use simpler datasets than those in Sec. 5.1. If the matrices $U_1, U_2'$ in Alg. 1 are dense, the final polynomials in $F$ tend to have many terms, leading to long input sequences. Thus, we construct datasets using $U_1, U_2'$ with a moderate density $\sigma \in [0, 1]$ so that the maximum sequence length becomes less than 4096. Specifically, we used $\sigma = 1.0, 0.6, 0.3, 0.2$ for $n = 2, 3, 4, 5$, respectively. The training set has one million samples, and the test set has one thousand samples. Other settings are the same as in Sec. 5.1 Using a more efficient attention mechanism such as (Kitaev et al., 2020; Ding et al., 2023; Sun et al., 2023) to handle larger polynomial sets will be a future work.

Tab. 2 shows that transformers trained on the constructed datasets compute correct $\prec_{\text{lex}}$-Gröbner bases with moderate accuracy. Nevertheless, even when the transformer cannot produce the correct basis, the *support*[8] of each polynomial in the predicted basis tends to be correct. Since there are efficient Gröbner basis computation algorithms if the support of each polynomial in Gröbner basis is accessible (Traverso, 1997), our experiments successfully show the transformer's potential in accelerating Gröbner computation. Lastly, the last two columns of Tab. 2 suggest that learning with $\prec_{\text{grvlex}}$ may not be as effective as that with $\prec_{\text{lex}}$.[9] In contrast, $\prec_{\text{grvlex}}$ is preferred in computational algebra because of its empirical runtime reduction in Gröbner basis computation. We consider that the well-structured nature of $\prec_{\text{lex}}$-Gröbner bases of ideals in shape position (c.f. Eq. (4.1)) may be favorable to transformers. Refer to App. D for success and failure examples. Particularly, the failure examples show that the incorrect outputs from transformers are still very reasonable.

## 6 DISCUSSION FROM AN ALGEBRAIC VIEWPOINT AND OPEN QUESTIONS

Without the Gröbner condition, Prob. 4.6 of characterizing the $A$ such that $F = AG$ and $\langle F \rangle = \langle G \rangle$ has been studied in (Busé et al., 2001; Busé, 2001) to provide an algebraic necessary and sufficient condition for the polynomial system of $F$ to have a solution outside the variety defined by $G$. This condition is expressed explicitly by multivariate resultants. However, strong additional assumptions are required: $A, F, G$ are homogeneous, $G$ is a regular sequence, and in the end, $\langle F \rangle = \langle G \rangle$ is only satisfied up to saturation. Thus, they are not compatible with our setting and method for Prob. 4.1.

Prop. 4.8 states that any matrix $A \in \text{SL}_n(k[x_1, \ldots, x_n])$ satisfies Prob. 4.6. This raises two sets of open questions: (i) *are there matrices outside* $\text{SL}_n(k[x_1, \ldots, x_n])$ *satisfying Prob. 4.6? Can we sample them?* and (ii) *is it possible to efficiently sample matrices of* $\text{SL}_n(k[x_1, \ldots, x_n])$? To efficiently generate our dataset, we have restricted ourselves to sampling matrices having a Bruhat decomposition (see Eq. (4.2)), which is a strict subset of $\text{SL}_n(k[x_1, \ldots, x_n])$. Sampling matrices in $\text{SL}_n(k[x_1, \ldots, x_n])$ remains an open question. Thanks to the Suslin's stability theorem and its algorithmic proofs (Suslin, 1977; Park & Woodburn, 1995; Lombardi & Yengui, 2005), $\text{SL}_n(k[x_1, \ldots, x_n])$ is generated by elementary matrices and a decomposition into a product of elementary matrices can be computed algorithmically. One may hope to use sampling of elementary matrices to sample matrices of $\text{SL}_n(k[x_1, \ldots, x_n])$. It is unclear whether this can be efficient as many elementary matrices are needed (Lombardi & Yengui, 2005).

In the experiments, we restricted ourselves to polynomial sets with a moderate sequence length so that a standard transformer can handle it. Algebraically, this relates to zero-dimensional ideals with small linear dimension (i.e., dimension of $k[x_1, \ldots, x_n]/\langle F \rangle$, see Def. A.4). To cover more general cases, zero-dimensional ideals with large linear dimensions and positive-dimensional ideals have to be further considered, which needs breakthroughs both in machine learning and algebra.

## 7 CONCLUSION

Solving polynomial systems, or computing Gröbner bases, are fundamental in various applications but known for its doubly exponential time complexity. In this study, we proposed the first training framework of transformers that realizes Gröbner basis computation via learning. The main technical challenge lies in an efficient generation of many pairs of (non-Gröbner set, Gröbner basis) to construct a training set, leading us to address two unexplored algebraic problems: random generation of Gröbner bases and the backward Gröbner problem. We resolved these problems in the case of zero-dimensional radical ideals, which are important in various applications. Our experiments validate that the proposed method achieves an extreme acceleration in dataset generation and that the transformer's ability in Gröbner basis computation. In a broad view, our study indicates that transformers may serve as a solver for NP-hard problems, while most prior studies only address rather easy/moderate-level problems. Besides, our results give a new motivation to computational algebra to consider and address several unexplored problems, posing a new direction and potential interactions of computational algebra and machine learning.

---

[8]The support of a polynomial $f$ refers to the set of terms in $f$.

[9]The training with $\prec_{\text{grvlex}}$ used a dataset that is essentially identical to $\mathcal{D}_2(\mathbb{F}_7)$. We simply apply the FGLM algorithm to $G$ and a term reordering to $F$ for each $(F, G) \in \mathcal{D}_2(\mathbb{F}_7)$.

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

## A   BASIC DEFINITIONS IN ALGEBRA

**Definition A.1** (Ring, Field (Atiyah & MacDonald (1994), Chap. 1 §1))**.** A set $R$ with an additive operation $+$ and a multiplicative operation $*$ is called a (commutative) ring if it satisfies the following conditions:

1.  $a + (b + c) = (a + b) + c$ for any $a, b, c \in R$,

2.  there exists $0 \in R$ such that $a + 0 = 0 + a = a$ for any $a \in R$,

3.  for any $a \in R$, there exists $-a$ such that $a + (-a) = (-a) + a = 0$,

4.  $a + b = b + a$ for any $a, b \in R$,

5.  $a * (b * c) = (a * b) * c$ for any $a, b, c \in R$,

6.  there exists $1 \in R$ such that $a * 1 = 1 * a = a$ for any $a \in R$,

7.  $a * (b + c) = a * b + a * c$ for any $a, b, c \in R$,

8. $(a + b) * c = a * c + b * c$ for any $a, b, c \in R$,

9. $a * b = b * a$ for any $a, b \in R$.

A commutative ring $R$ is called a field if it satisfies the following condition

10. for any $a \in R \setminus \{0\}$, there exists $a^{-1}$ such that $a * a^{-1} = a^{-1} * a = 1$.

**Definition A.2** (Polynomial Ring (Atiyah & MacDonald (1994), Chap. 1 §1))**.** In Definition A.1, $k[x_1, \ldots, x_n]$, the set of all $n$-variate polynomials with coefficients in $k$, satisfies all conditions (1)-(9). Thus, $k[x_1, \ldots, x_n]$ is called a polynomial ring.

**Definition A.3** (Quotient Ring (Atiyah & MacDonald (1994), Chap. 1 §1))**.** Let $R$ be a ring and $I$ an ideal of $R$. For each $f \in R$, we set $[f] = \{g \in R \mid f - g \in I\}$. Then, the set $\{[f] \mid f \in R\}$ is called the quotient ring of $R$ modulo $I$ and denoted by $R/I$. Indeed, $R/I$ is a ring with an additive operation $+$ and a multiplicative operation $*$, where $[f] + [g] = [f + g]$ and $[f] * [g] = [f * g]$ for $f, g \in R$ respectively.

**Definition A.4** (Zero-dimensional ideal (Cox et al. (2015), Chap. 5 §3, Thm. 6))**.** Let $F$ be a set of polynomials in $k[x_1, \ldots, x_n]$. An ideal $\langle F \rangle$ is called a *zero-dimensional ideal* if the $k$-linear space $k[x_1, \ldots, x_n]/\langle F \rangle$ is finite-dimensional, where $k[x_1, \ldots, x_n]/\langle F \rangle$ is the quotient ring of $k[x_1, \ldots, x_n]$ modulo $\langle F \rangle$.

**Definition A.5** (Radical ideal (Atiyah & MacDonald (1994), Chap. 1 §1))**.** For an ideal $I$ of $k[x_1, \ldots, x_n]$, the set $\{f \in k[x_1, \ldots, x_n] \mid f^m \in I$ for a positive integer $m\}$ is called the radical of $I$ and denoted by $\sqrt{I}$. Also, $I$ is called a radical ideal if $I = \sqrt{I}$.

**Definition A.6** (Syzygy (Becker et al. (1993), Chap. 3, §3))**.** Let $F = \{f_1, \ldots, f_s\} \subset k[x_1, \ldots, x_n]$. A *syzygy* of $F$ is an $s$-tuple of polynomials $(q_1, \ldots, q_s) \in k[x_1, \ldots, x_n]^s$ such that $q_1 f_1 + \cdots + q_s f_s = 0$.

**Definition A.7** (Term (Becker et al. (1993), Chap. 2, §1))**.** For a polynomial $f = \sum_{\alpha_1, \ldots, \alpha_n} c_{\alpha_1, \ldots, \alpha_n} x_1^{\alpha_1} \cdots x_n^{\alpha_n}$ with $c_{\alpha_1, \ldots, \alpha_n} \in K$ and $\alpha_1, \ldots, \alpha_n \in \mathbb{Z}_{\geq 0}$, each $x_1^{\alpha_1} \cdots x_n^{\alpha_n}$ is called a term in $f$.

**Definition A.8** (Total Degree (Cox et al. (2015), Chap. 1 §1, Def. 3))**.** For a term $x_1^{\alpha_1} \cdots x_n^{\alpha_n}$, its total degree is the sum of indices $\alpha_1 + \cdots + \alpha_n$. For a polynomial $f$, the total degree of $f$ is the maximal total degree of terms in $f$.

**Definition A.9** (Term order (Becker et al. (1993), Definition 5.3))**.** A *term order* $\prec$ is a relation between terms such that

1. (comparability) for different terms $x_1^{\alpha_1} \cdots x_n^{\alpha_n}$ and $x_1^{\beta_1} \cdots x_n^{\beta_n}$, either $x_1^{\alpha_1} \cdots x_n^{\alpha_n} \prec x_1^{\beta_1} \cdots x_n^{\beta_n}$ or $x_1^{\beta_1} \cdots x_n^{\beta_n} \prec x_1^{\alpha_1} \cdots x_n^{\alpha_n}$ holds,

2. (order-preserving) for terms $x_1^{\alpha_1} \cdots x_n^{\alpha_n}, x_1^{\beta_1} \cdots x_n^{\beta_n}$ and $x_1^{\gamma_1} \cdots x_n^{\gamma_n} \neq 1$, if $x_1^{\alpha_1} \cdots x_n^{\alpha_n} \prec x_1^{\beta_1} \cdots x_n^{\beta_n}$ then $x_1^{\alpha_1 + \gamma_1} \cdots x_n^{\alpha_n + \gamma_n} \prec x_1^{\beta_1 + + \gamma_1} \cdots x_n^{\beta_n + \gamma_n}$ holds,

3. (minimality of 1) the term 1 is the smallest term i.e. $1 \prec x_1^{\alpha_1} \cdots x_n^{\alpha_n}$ for any term $x_1^{\alpha_1} \cdots x_n^{\alpha_n} \neq 1$.

**Example A.10.** The *graded lexicographic order* $\prec_{\mathrm{grlex}}$ prioritizes terms with higher total degree.[10] For tie-break, the lexicographic order is used, e.g.,

$$1 \prec_{\mathrm{grlex}} x_n \quad \text{and} \quad x_2 \prec_{\mathrm{grlex}} x_3^2 \quad \text{and} \quad x_1 x_2 x_3^2 \prec_{\mathrm{grlex}} x_1 x_2^2 x_3. \tag{A.1}$$

Term orders prioritizing lower total degree terms as $\prec_{\mathrm{grlex}}$ are called graded term orders.

# B BUCHBERGER–MÖLLER ALG. FOR PROB. 4.1

Here, we discuss another approach for Prob. 4.1 using the Buchberger–Möller (BM) algorithm (Möller & Buchberger, 1982). Although we did not adopt this approach, we include this

---

[10]The total degree of term $x_1^{\alpha_1} \cdots x_n^{\alpha_n}$ refers to $\sum_{i=1}^{n} \alpha_i$. The total degree of polynomial $f$ refers to the maximum total degree of the terms in $f$.

for completeness because many algorithm variants have been recently developed and applied extensively in machine learning and other data-centric applications.

Given a set of points $\mathbb{X} \subset k^n$ and a graded term order, the BM algorithm computes a Gröbner basis of its vanishing ideal $I(\mathbb{X}) = \{g \in k[x_1, \ldots, x_n] \mid g(p) = 0, \forall p \in \mathbb{X}\}$. While several variants follow in computational algebra (Kehrein & Kreuzer, 2006; Abbott et al., 2008; Heldt et al., 2009; Fassino, 2010; Limbeck, 2013; Kera, 2022), interestingly, it is also recently tailored for machine learning (Livni et al., 2013; Király et al., 2014; Hou et al., 2016; Kera & Hasegawa, 2018; Kera & Hasegawa, 2019; Kera & Hasegawa, 2020; Wirth & Pokutta, 2022; Wirth et al., 2023) and has been applied to various contexts such as machine learning (Shao et al., 2016; Yan et al., 2018), signal processing (Wang & Ohtsuki, 2018; Wang et al., 2019; Wang & Deng, 2020), nonlinear dynamics (Kera & Hasegawa, 2016; Karimov et al., 2020), and more (Kera & Iba, 2016; Iraji & Chitsaz, 2017; Antonova et al., 2020). Such a wide range of applications derives from the distinguishing design of the BM algorithm: while most computer-algebraic algorithms take a set of polynomials as input, it takes a set of points (i.e., dataset).

Therefore, to address Prob. 4.1, one may consider using the BM algorithm or its variants, e.g., by running the BM algorithm $m$ times while sampling diverse sets of points. An important caveat is that Gröbner bases that can be given by the BM algorithm may be more restrictive than those considered in the main text (i.e., the Gröbner bases of ideals in shape position). For example, the former generates the largest ideals that have given $k$-rational points for their roots, whereas this is not the case for the latter. Another drawback of using the BM algorithm is its large computational cost. The time complexity of the BM algorithm is $\mathcal{O}(n \cdot |\mathbb{X}|^3)$. Furthermore, we need $\mathcal{O}(n^d)$ points to obtain a Gröbner basis that includes a polynomial of degree $d$ in the average case. Therefore, the BM algorithm does not fit our settings that a large number of Gröbner bases are needed (i.e., $m \approx 10^6$). Accelerating the BM algorithm by reusing the results of runs instead of independently running the algorithm many times can be interesting for future work.

## C  PROOFS

**Theorem 4.7.** *Let $G = (g_1, \ldots, g_t)^\top$ be a Gröbner basis of a zero-dimensional ideal in $k[x_1, \ldots, x_n]$. Let $F = (f_1, \ldots, f_s)^\top = AG$ with $A \in k[x_1, \ldots, x_n]^{s \times t}$.*

1. *If $\langle F \rangle = \langle G \rangle$, it implies $s \geq n$.*

2. *If $A$ has a left-inverse in $k[x_1, \ldots, x_n]^{s \times t}$, $\langle F \rangle = \langle G \rangle$ holds.*

3. *The equality $\langle F \rangle = \langle G \rangle$ holds if and only if there exists a matrix $B \in k[x_1, \ldots, x_n]^{t \times s}$ such that each row of $BA - E_t$ is a syzygy[11] of $G$, where $E_t$ is the identity matrix of size $t$.*

*Proof.*
(1) In general, if an ideal $I$ is generated by $s$ elements and $s < n$, then the Krull dimension of $k[x_1, \ldots, x_n]/I$ satisfies that $\dim k[x_1, \ldots, x_n]/I \geq n - s > 0$ (Krull's principal ideal theorem (Eisenbud, 2013, §10)). Since the Krull dimension of $k[x_1, \ldots, x_n]/\langle G \rangle$ is 0, we have $s \geq n$.

(2) From $F = AG$, we have $\langle F \rangle \subset \langle G \rangle$. If $A$ has a left-inverse $B \in k[x_1, \ldots, x_n]^{t \times s}$, we have $BF = BAG = G$, indicating $\langle F \rangle \supset \langle G \rangle$. Therefore, we have $\langle F \rangle = \langle G \rangle$.

(3) If the equality $\langle F \rangle = \langle G \rangle$ holds, then there exists a $t \times s$ matrix $B \in k[x_1, \ldots, x_n]^{t \times s}$ such that $G = BF$. Since $F$ is defined as $F = AG$, we have $G = BF = BAG$ and $G = E_t G$ in $k[x_1, \ldots, x_n]^t$. Therefore we obtain $(BA - E_t)G = 0$. In particular, each row of $BA - E_t$ is a syzygy of $G$. Conversely, if there exists a $t \times s$ matrix $B \in k[x_1, \ldots, x_n]^{t \times s}$ such that each row of $BA - E_t$ is a syzygy of $G$, then we have $(BA - E_t)G = 0$ in $k[x_1, \ldots, x_n]^t$, therefore the equality $\langle F \rangle = \langle G \rangle$ holds since we have $G = E_t G = BAG = BF$. $\square$

**Proposition 4.8.** *For any $A \in k[x_1, \ldots, x_n]^{n \times n}$ with $\det(A) \in k \setminus \{0\}$, we have $\langle F \rangle = \langle G \rangle$.*

---

[11]Refer to App. A for the definition.

*Proof.* From the Cramer's rule, there exists $B \in k[x_1, \ldots, x_n]^{n \times n}$ such that $BA = \det(A)E_n$, where $E_n$ denotes the $n$-by-$n$ identity matrix. Indeed, the $i$-th row $B_i$ of $B$ satisfies for $i = 1, \ldots, n$,

$$B_i = \left( \det\left( \tilde{A}_1^{(i)} \right), \ldots, \det\left( \tilde{A}_n^{(i)} \right) \right), \tag{C.1}$$

where $\tilde{A}_j^{(i)}$ is the matrix $A$ with the $j$-th column replaced by the $i$-th canonical basis $e_i = (0, \ldots, 1, \ldots, 0)^\top$. Since $\det(A)$ is a non-zero constant, $A$ has the left-inverse $B/\det(A)$ in $k[x_1, \ldots, x_n]$. Thus $\langle F \rangle = \langle G \rangle$ from Thm. 4.7. $\qquad\square$

**Theorem 4.9.** *Consider polynomial ring $k[x_1, \ldots, x_n]$. Given dataset size $m$, maximum degrees $d, d'$, maximum size of non-Gröbner set $s_{\max} \geq n$, and term order $\prec$, Alg. 1 returns a collection $\mathcal{D} = \{(F_i, G_i)\}_{i=1}^m$ with the following properties: For all $i = 1, \ldots, m$,*

1. *Both $F_i, G_i \subset k[x_1, \ldots, x_n]$ are finite sets and $|F_i| \leq s_{\max}$.*

2. *The set $G_i$ is a $\prec$-Gröbner basis. The set $F_i$ is not, unless $G_i, U_1, U_2', P$ are sampled in a non-trivial Zariski closed subset.*[12]

3. *The ideal $\langle F_i \rangle = \langle G_i \rangle$ is a zero-dimensional ideal in shape position.*

*The time complexity is $\mathcal{O}(m(nS_{1,d} + s^2 S_{n,d'} + (n^2 + s^2)M_{n,2d'+d}))$ when $\prec = \prec_{\mathrm{lex}}$, where $S_{n,d}$ denotes the complexity of sampling an $n$-variate polynomial with total degree at most $d$, and $M_{n,d}$ denotes that of multiplying two $n$-variate polynomials with total degree at most $d$. If $\prec \neq \prec_{\mathrm{lex}}$, $\mathcal{O}(nd^3)$ is additionally needed.*

*Proof.* Outside of the Zariski subset part, statements 1–3 are trivial from Alg. 1 and the discussion in Sec.s 4.1 and 4.2. To obtain the desired Zariski subsets, we consider the vector space of polynomials of degree $d + 2d'$ or less. We remark that if $F_i$ is a $\prec$-Gröbner basis, its leading terms must belong to a finite amount of possibilities. For a polynomial to have a given term as its leading term, zero conditions on terms greater than this term are needed, defining a closed Zariski subset condition. By considering the finite union of all these conditions, we obtain the desired result.

To obtain one pair $(F, G)$, the random generation of $G$ needs $\mathcal{O}(nS_{1,d})$, and the backward transform from $G$ to $F$ needs $\mathcal{O}(s^2 S_{n,d'})$ to get $U_1, U_2$ and $(n^2 + s^2)M_{n,2d'+d}$ for the multiplication $F = U_1 P U_2 G$. Note that the maximum total degree of polynomials in $F$ is $2d' + d$. $\qquad\square$

## D  TRAINING SETUP AND ADDITIONAL EXPERIMENTAL RESULTS.

This section provides the supplemental information of our experiments presented in Sec. 5.

### D.1  GRÖBNER BASIS COMPUTATION ALGORITHMS

In Tab. 1, we tested three algorithms provided in Sagemath with the libSingular backend for forward generation.

**STD (`libsingular:std`):**   The standard Buchberger algorithm.

**SLIMGB (`libsingular:slimgb`):**   A variant of the Faugère's F4 algorithm. Refer to (Brickenstein, 2010).

**STDFGLM (`libsingular:stdfglm`):**   Fast computation using STD with the graded reverse lexicographic order followed by the FGLM for the change of term orders.

### D.2  DATASET PROFILES

We used datasets generated with a density control for training a transformer. The runtime comparison for these datasets is given in Tab. 3. Because of the density control, the forward generation uses

---

[12]This can happen with probability zero if $k$ is infinite and very low probability over large finite field.

Table 3: Runtime comparison (in seconds/millisecond) of forward generation (F.) and backward generation (B.) of dataset $\mathcal{D}_n(\mathbb{F}_p)$ of size 1000. The forward generation used either of the three algorithms provided in Sagemath with the libSingular backend. The backward generation is two orders of magnitude faster than the forward generation.

| Method | $\mathbb{F}_7[x_1,\dots,x_n]$ | | | | $\mathbb{F}_{31}[x_1,\dots,x_n]$ | | | |
|---|---|---|---|---|---|---|---|---|
| | $n=2$ | $n=3$ | $n=4$ | $n=5$ | $n=2$ | $n=3$ | $n=4$ | $n=5$ |
| | $\sigma=1$ | 0.6 | 0.3 | 0.2 | $\sigma=1$ | 0.6 | 0.3 | 0.2 |
| F. (STD) [sec] | 0.37 | 0.40 | 6.33 | 4.29 | 0.38 | 0.42 | 7.32 | 4.77 |
| F. (SLIMGB) | 0.44 | 0.58 | 5.98 | 5.79 | 0.41 | 0.61 | 5.79 | 6.39 |
| F. (STDFGLM) | 0.99 | 1.11 | 6.24 | 6.34 | 1.01 | 1.11 | 6.34 | 6.62 |
| B. (ours) [msec] | **2.23** | **2.47** | **2.51** | **2.78** | **2.12** | **2.45** | **2.42** | **2.79** |

Table 4: A profile of the generated datasets (relevant to Tab. 1). The standard deviation is shown in the superscript.

| Method | $\mathbb{F}_7[x_1,\dots,x_n]$ | | | | $\mathbb{F}_{31}[x_1,\dots,x_n]$ | | | |
|---|---|---|---|---|---|---|---|---|
| | $n=2$ | $n=3$ | $n=4$ | $n=5$ | $n=2$ | $n=3$ | $n=4$ | $n=5$ |
| size of $F$ | $3.0^{(\pm0.8)}$ | $4.0^{(\pm0.7)}$ | $5.0^{(\pm0.8)}$ | $6.0^{(\pm0.8)}$ | $3.0^{(\pm0.8)}$ | $3.9^{(\pm0.8)}$ | $4.9^{(\pm0.8)}$ | $6.0^{(\pm0.8)}$ |
| max deg in $F$ | $8.9^{(\pm1.8)}$ | $9.8^{(\pm1.2)}$ | $10^{(\pm1.0)}$ | $10^{(\pm0.8)}$ | $8.7^{(\pm1.8)}$ | $9.9^{(\pm1.3)}$ | $10^{(\pm0)}$ | $10^{(\pm0)}$ |
| min deg in $F$ | $5.6^{(\pm1.8)}$ | $5.9^{(\pm1.9)}$ | $6.3^{(\pm1.9)}$ | $6.5^{(\pm1.8)}$ | $5.4^{(\pm1.9)}$ | $6.1^{(\pm1.8)}$ | $6.4^{(\pm1.8)}$ | $6.6^{(\pm1.6)}$ |
| #terms in $F$ | $23^{(\pm9.6)}$ | $34^{(\pm10)}$ | $45^{(\pm10)}$ | $57^{(\pm11)}$ | $22^{(\pm9.5)}$ | $33^{(\pm10)}$ | $45^{(\pm10)}$ | $57^{(\pm11)}$ |
| GB ratio [%] | 0.10 | 0.00 | 0.00 | 0.00 | 0.20 | 0.00 | 0.00 | 0.00 |
| size of $G$ | $2.0^{(\pm0)}$ | $3.0^{(\pm0)}$ | $4.0^{(\pm0)}$ | $5.0^{(\pm0)}$ | $2.0^{(\pm0)}$ | $3.0^{(\pm0)}$ | $4.0^{(\pm0)}$ | $5.0^{(\pm0)}$ |
| max deg in $G$ | $4.3^{(\pm1.0)}$ | $4.3^{(\pm0.8)}$ | $4.4^{(\pm0.8)}$ | $4.4^{(\pm0.7)}$ | $4.2^{(\pm1.0)}$ | $4.4^{(\pm0.8)}$ | $4.4^{(\pm0.7)}$ | $4.5^{(\pm0.70)}$ |
| min deg in $G$ | $3.7^{(\pm1.5)}$ | $3.2^{(\pm1.6)}$ | $3.0^{(\pm1.7)}$ | $2.8^{(\pm1.6)}$ | $3.4^{(\pm1.6)}$ | $3.6^{(\pm1.6)}$ | $3.5^{(\pm1.7)}$ | $3.4^{(\pm1.7)}$ |
| #terms in $G$ | $8.1^{(\pm2.4)}$ | $11^{(\pm3.6)}$ | $15^{(\pm4.9)}$ | $19^{(\pm6.1)}$ | $7.7^{(\pm2.5)}$ | $12^{(\pm3.6)}$ | $16^{(\pm4.8)}$ | $20^{(\pm6.1)}$ |
| GB ratio [%] | 100 | 100 | 100 | 100 | 100 | 100 | 100 | 100 |

Table 5: A profile of datasets generated with a density control (relevant to Tab. 3). The standard deviation is shown in the superscript.

| Method | $\mathbb{F}_7[x_1,\dots,x_n]$ | | | | $\mathbb{F}_{31}[x_1,\dots,x_n]$ | | | |
|---|---|---|---|---|---|---|---|---|
| | $n=2$ | $n=3$ | $n=4$ | $n=5$ | $n=2$ | $n=3$ | $n=4$ | $n=5$ |
| | $\sigma=1$ | 0.6 | 0.3 | 0.2 | $\sigma=1$ | 0.6 | 0.3 | 0.2 |
| size of $F$ | $3.0^{(\pm0.8)}$ | $4.0^{(\pm0.8)}$ | $5.0^{(\pm0.8)}$ | $6.0^{(\pm0.8)}$ | $3.0^{(\pm0.8)}$ | $4.0^{(\pm0.8)}$ | $5.0^{(\pm0.8)}$ | $6.0^{(\pm0.8)}$ |
| max deg in $F$ | $8.8^{(\pm1.9)}$ | $8.5^{(\pm1.9)}$ | $8.4^{(\pm1.9)}$ | $8.6^{(\pm1.8)}$ | $8.7^{(\pm1.9)}$ | $8.8^{(\pm1.9)}$ | $8.6^{(\pm1.8)}$ | $8.7^{(\pm1.8)}$ |
| min deg in $F$ | $5.6^{(\pm1.9)}$ | $4.1^{(\pm2.3)}$ | $3^{(\pm2.3)}$ | $2.6^{(\pm2.3)}$ | $5.5^{(\pm1.9)}$ | $4.3^{(\pm2.2)}$ | $3.2^{(\pm2.4)}$ | $2.9^{(\pm2.4)}$ |
| #terms in $F$ | $23^{(\pm9.6)}$ | $26^{(\pm10)}$ | $29^{(\pm10)}$ | $34^{(\pm11)}$ | $23^{(\pm9.5)}$ | $27^{(\pm10)}$ | $30^{(\pm10)}$ | $35^{(\pm11)}$ |
| GB ratio [%] | 0.10 | 1.60 | 2.20 | 0.70 | 0.10 | 1.50 | 1.80 | 1.10 |
| size of $G$ | $2.0^{(\pm0)}$ | $3.0^{(\pm0)}$ | $4.0^{(\pm0)}$ | $5.0^{(\pm0)}$ | $2.0^{(\pm0)}$ | $3.0^{(\pm0)}$ | $4.0^{(\pm0)}$ | $5.0^{(\pm0)}$ |
| max deg in $G$ | $4.3^{(\pm1)}$ | $4.4^{(\pm0.8)}$ | $4.4^{(\pm0.79)}$ | $4.5^{(\pm0.7)}$ | $4.2^{(\pm1.1)}$ | $4.4^{(\pm0.8)}$ | $4.5^{(\pm0.7)}$ | $4.5^{(\pm0.7)}$ |
| min deg in $G$ | $3.8^{(\pm1.6)}$ | $3.2^{(\pm1.7)}$ | $3^{(\pm1.7)}$ | $2.9^{(\pm1.7)}$ | $3.5^{(\pm1.6)}$ | $3.6^{(\pm1.7)}$ | $3.5^{(\pm1.7)}$ | $3.5^{(\pm1.8)}$ |
| #terms in $G$ | $8.1^{(\pm2.5)}$ | $12^{(\pm3.6)}$ | $15^{(\pm4.9)}$ | $19^{(\pm6.1)}$ | $7.7^{(\pm2.5)}$ | $12^{(\pm3.7)}$ | $16^{(\pm4.9)}$ | $20^{(\pm6.1)}$ |
| GB ratio [%] | 100 | 100 | 100 | 100 | 100 | 100 | 100 | 100 |

less runtime but still needs a long runtime if one needs to construct a training set with a million samples. The backward generation used roughly 100 times less runtime. The dataset profile of datasets are given in Tabs. 4 and 5.

### D.3 TRAINING OF TRANSFORMERS.

To examine the transformer's ability to learn Gröbner basis computation, we focus on a standard architecture and training setup. We used a transformer model (Vaswani et al., 2017) with standard

Table 6: Success examples ($n = 2$)

| ID | $F$ | $G$ |
|----|-----|-----|
| 15 | $f_1 = -2x_0^2x_1^4 - x_0^2x_1 - 2x_0x_1 - x_1^4 + 3x_1^3$ 
 $f_2 = -2x_0^3x_1^4 - x_0^3x_1 - 2x_0^2x_1 - x_0x_1^4 +$ 
 $3x_0x_1^3 + x_0x_1^2 - 3x_1^5 + 2x_1^4 - 2x_1^3 - 1$ 
 $f_3 = x_0^3x_1^6 - 3x_0^3x_1^3 - x_0^2x_1^3 + 3x_0x_1^6 - 2x_0x_1^5 -$ 
 $3x_0x_1^4 - x_1^4 - 3x_1^3 - 3$ 
 $f_4 = -3x_0^3x_1^3 - 2x_0^2x_1^6 + x_0^2x_1^5 - x_0^2x_1^4 -$ 
 $2x_0^2x_1^3 + x_0^2x_1^2 + 3x_0^2x_1 - 3x_0x_1^5 - 2x_0x_1^4 +$ 
 $3x_0x_1^3 - 2x_0x_1 + x_0 - 2x_1^6 - x_1^5 - 3x_1^3 + 2x_1^2$ | $g_1 = x_0 - 3x_1^3 + 2x_1^2$ 
 $g_2 = -2x_1^3 - 1$ |
| 16 | $f_1 = -x_0^2x_1^2 - x_0^2x_1 - 3x_0x_1^3 - 3x_1^7 - 2x_1^6 + x_1^3$ 
 $f_2 = -x_0^4x_1^3 - x_0^4x_1^2 - 3x_0^3x_1^4 - 3x_0^2x_1^8 -$ 
 $2x_0^2x_1^7 + x_0^2x_1^4 - x_0x_1^3 - x_1^7 - 3x_1^6 - 2x_1^3 +$ 
 $x_1 + 1$ 
 $f_3 = -x_0^4x_1^3 - x_0^4x_1^2 - 3x_0^3x_1^4 - 3x_0^2x_1^8 -$ 
 $2x_0^2x_1^7 + x_0^2x_1^4 - 2x_0^2x_1^3 - 2x_0x_1^7 + x_0x_1^6 +$ 
 $3x_0x_1^3 + 2x_0x_1 + 3x_0 + x_1^4 + 3x_1^3 + 2$ | $g_1 = x_0 + x_1^4 + 3x_1^3 + 2$ 
 $g_2 = x_1 + 1$ |
| 19 | $f_1 = x_0 + 3x_1^5 + 3x_1^2 + 2x_1 - 3$ 
 $f_2 = -2x_0^2x_1^2 - 2x_0^2x_1 + 2x_0^2 + x_0x_1^7 + x_0x_1^6 -$ 
 $x_0x_1^5 + x_0x_1^4 - 3x_0x_1^3 + x_0x_1^2 + 3x_0x_1 +$ 
 $x_0 - 2x_1^5 - 2x_1^4 - 2x_1^3 + 3x_1^2$ 
 $f_3 = 3x_0^3x_1^2 + 3x_0^3x_1 - 2x_0^3 + 2x_0^2x_1^7 + 2x_0^2x_1^6 -$ 
 $x_0^2x_1^5 - x_0^2x_1^3 + x_0^2x_1^2 - 3x_0^2x_1 - 3x_0^2 - x_0x_1^6 -$ 
 $x_0x_1^5 - 2x_0x_1^4 - 3x_0x_1^3 + x_0x_1 - x_0 + x_1^7 +$ 
 $2x_1^5 + 3x_1^4 - 2x_1^3 + 3x_1^2$ | $g_1 = x_0 + 3x_1^5 + 3x_1^2 + 2x_1 - 3$ 
 $g_2 = -2x_1^5 - 2x_1^4 - 2x_1^3 + 3x_1^2$ |
| 23 | $f_1 = x_0 - x_1^5 - x_1^3 + 3x_1^2 + 2x_1$ 
 $f_2 = x_0^2x_1^2 - 3x_0^2 - x_0x_1^7 + 2x_0x_1^5 + 3x_0x_1^4 -$ 
 $2x_0x_1^3 - 2x_0x_1^2 + x_0x_1 + 2x_1^5 + 2x_1^3 + 2$ | $g_1 = x_0 - x_1^5 - x_1^3 + 3x_1^2 + 2x_1$ 
 $g_2 = 2x_1^5 + 2x_1^3 + 2$ |

architectures, e.g., 6 encoder/decoder layers, 8 attention heads, token embedding dimension of 512 dimensions, and feed-forward networks with 2048 inner dimensions. The dropout rate was set to 0.1. We used the AdamW optimizer (Loshchilov & Hutter, 2019) with $(\beta_1, \beta_2) = (0.9, 0.999)$ with no weight decay. The learning rate was initially set to $10^{-4}$ and then linearly decayed over training steps. All training samples are visited in a single epoch, and the total number of epochs was set to 10. The batch size was set to 8. At the inference time, output sequences are generated using a beam search with width 1.

Tabs. 6– 9 show examples of success cases and Tab. 10 shows examples of failure cases. One can see from the former that transformers accomplish difficult computations and from the latter, interestingly, that the incorrect predictions appear reasonable.

Table 7: Success examples ($n = 3$)

| ID | $F$ | $G$ |
|---|---|---|
| 5 | $f_1 = x_1 - x_2^4$ 
 $f_2 = 2x_0^3 x_1 - x_0^2 x_1 - 2x_2^2$ 
 $f_3 = 2x_0^3 x_1 + 2x_0^2 x_1 - 3x_0^2 x_2^4 + x_0 - 2x_2^2 + 3$ | $g_1 = x_0 + 3$ 
 $g_2 = x_1 - x_2^4$ 
 $g_3 = -2x_2^2$ |
| 6 | $f_1 = 0$ 
 $f_2 = x_0 + 2x_2^5 - 2x_2^4 + x_2^3 - 2$ 
 $f_3 = -x_0^2 x_1 x_2 - 3x_0^2 - 2x_0 x_1 x_2^6 + 2x_0 x_1 x_2^5 - x_0 x_1 x_2^4 + 2x_0 x_1 x_2 + x_0 x_2^5 - x_0 x_2^4 - 3x_0 x_2^3 - x_0 + 3x_2^5 + 2x_2^4 - 2x_2^3$ 
 $f_4 = x_0^3 x_1 x_2^2 + 3x_0^3 x_2 - x_0^2 x_1^2 x_2^2 + 2x_0^2 x_1 x_2^7 - 2x_0^2 x_1 x_2^6 + x_0^2 x_1 x_2^5 - 2x_0^2 x_1 x_2^2 - 3x_0^2 x_1 x_2 - x_0^2 x_2^6 + x_0^2 x_2^5 + 3x_0^2 x_2^4 + x_0^2 x_2 - 2x_0 x_1^2 x_2^7 + 2x_0 x_1^2 x_2^6 - x_0 x_1^2 x_2^5 + 2x_0 x_1^2 x_2^2 - 2x_0 x_1^2 x_2 + 2x_0 x_1^2 + x_0 x_1 x_2^6 - x_0 x_1 x_2^5 - 3x_0 x_1 x_2^4 - x_0 x_1 x_2 - 3x_0 x_2^6 - 2x_0 x_2^5 + 2x_0 x_2^4 + 3x_1^2 x_2^6 + x_1^2 x_2^5 + x_1^2 x_2^4 + 2x_1^2 x_2^3 - 3x_1^2 x_2 + 3x_1^2 + 3x_1 x_2^6 + 2x_1 x_2^5 - 2x_1 x_2^4 + x_1 - x_2^5 + 2x_2^3 - 3x_2^2 - 3x_2$ | $g_1 = x_0 + 2x_2^5 - 2x_2^4 + x_2^3 - 2$ 
 $g_2 = x_1 - x_2^5 + 2x_2^3 - 3x_2^2 - 3x_2$ 
 $g_3 = 3x_2^5 + 2x_2^4 - 2x_2^3$ |
| 8 | $f_1 = x_1 + 2x_2^5 + 2x_2^2 + 2x_2 + 3$ 
 $f_2 = -3x_0^2 x_2^2 + 2x_0 x_1 x_2 + 2x_0 x_2^7 - x_0 x_2^6 + 2x_0 x_2^5 + 3x_0 x_2^4 + 2x_0 x_2^3 + x_1 x_2^6 + 3x_1 x_2^5 + x_1 x_2^4 - 2x_1 x_2^3 + x_1 x_2^2 - 2x_2^5 - x_2^4 + 2x_2^2 - 3$ 
 $f_3 = 3x_0^2 x_1 x_2^2 + x_0^2 x_2^4 - 2x_0 x_1^2 x_2 - 2x_0 x_1 x_2^7 + x_0 x_1 x_2^6 - 2x_0 x_1 x_2^5 - 3x_0 x_1 x_2^4 + 2x_0 x_1 x_2^3 - 3x_0 x_2^9 - 2x_0 x_2^8 - 3x_0 x_2^7 - x_0 x_2^6 - 3x_0 x_2^5 - x_1^2 x_2^6 - 3x_1^2 x_2^5 - x_1^2 x_2^4 + 2x_1^2 x_2^3 - x_1^2 x_2^2 + 2x_1 x_2^8 - x_1 x_2^7 + 2x_1 x_2^6 + 3x_1 x_2^5 + 2x_1 x_2^4 + 3x_2^7 - 2x_2^6 - 3x_2^4 + x_2^2$ 
 $f_4 = x_0^2 x_1 x_2 + 2x_0^2 x_2^6 + 2x_0^2 x_2^3 + 2x_0^2 x_2^2 + 3x_0^2 x_2 + 2x_1^4 - x_1^3 x_2^5 + x_1^3 x_2^4 + 2x_1^3 x_2^2 + 2x_1^3 - x_1^2 x_2^6 - x_1^2 x_2^3 - x_1^2 x_2^2 + 2x_1^2 x_2$ 
 $f_5 = x_0 - 3x_2^5 - 2x_2^4 - 3x_2^3 - x_2^2 - 3x_2$ | $g_1 = x_0 - 3x_2^5 - 2x_2^4 - 3x_2^3 - x_2^2 - 3x_2$ 
 $g_2 = x_1 + 2x_2^5 + 2x_2^2 + 2x_2 + 3$ 

 $g_3 = -2x_2^5 - x_2^4 + 2x_2^2 - 3$ |
| 10 | $f_1 = -2x_1 x_2^6 - 3x_1 x_2^5 - x_1 x_2^4 + 3x_1 x_2^3 - 2x_1 x_2 + 3x_2^5 + x_2^4 - 2x_2^3 - x_2^2 + 3$ 
 $f_2 = 3x_0^2 x_1^2 x_2^6 + x_0^2 x_1^2 x_2^5 - 2x_0^2 x_1^2 x_2^4 - x_0^2 x_1^2 x_2^3 + 3x_0^2 x_1^2 x_2 - x_0^2 x_1 x_2^5 + 2x_0^2 x_1 x_2^4 + 3x_0^2 x_1 x_2^3 - 2x_0^2 x_1 x_2^2 - x_0^2 x_1 - 2x_0 x_1 x_2^8 - 3x_0 x_1 x_2^7 - x_0 x_1 x_2^6 + 3x_0 x_1 x_2^5 - 2x_0 x_1 x_2^3 + 3x_0 x_2^7 + x_0 x_2^6 - 2x_0 x_2^5 - x_0 x_2^4 + 3x_0 x_2^2 + x_0 - 3x_2^4 - 3x_2^3 - 2x_2 - 3$ 
 $f_3 = 0$ 
 $f_4 = -x_0 x_1 x_2 + x_0 x_2^6 + 3x_0 x_2^4 + x_0 x_2^3 + 3x_0 x_2^2 + 2x_1^2 x_2^2 - 2x_1 x_2^7 + x_1 x_2^5 - 2x_1 x_2^4 + x_1 x_2^3 - 3x_2^5 - x_2^4 + 2x_2^3 + x_2^2 - 3$ 
 $f_5 = x_0^3 x_1 x_2^2 - x_0^3 x_2^7 - 3x_0^3 x_2^5 - x_0^3 x_2^4 - 3x_0^3 x_2^3 - 2x_0^2 x_1^2 x_2^3 - 2x_0^2 x_1^2 x_2^2 + 2x_0^2 x_1 x_2^8 + 2x_0^2 x_1 x_2^7 - x_0^2 x_1 x_2^6 + x_0^2 x_1 x_2^5 + x_0^2 x_1 x_2^4 - x_0^2 x_1 x_2^3 + 3x_0^2 x_2^6 + x_0^2 x_2^5 - 2x_0^2 x_2^4 - x_0^2 x_2^3 + 3x_0^2 x_2 - 3x_0 x_1^3 x_2^3 + 3x_0 x_1^2 x_2^8 + 2x_0 x_1^2 x_2^6 + 3x_0 x_1^2 x_2^5 + 2x_0 x_1^2 x_2^4 + x_0 x_1 x_2^6 - 2x_0 x_1 x_2^5 - 3x_0 x_1 x_2^4 + 2x_0 x_1 x_2^3 + x_0 x_1 x_2 - 3x_0 x_2^7 + 3x_0 + 2x_1 x_2^7 - x_1 x_2^6 + 2x_1 x_2^5 + 2x_1 x_2^4 - x_1 x_2^3 + 2x_1 x_2^2 + 3x_1 x_2 + x_1 - x_2^6 - x_2^5 + 2x_2^4 - 2x_2^3 - x_2^2 + 2x_2 - 3$ | $g_1 = x_0 - 3x_2^4 - 3x_2^3 - 2x_2 - 3$ 

 $g_2 = x_1 - x_2^5 - 3x_2^3 - x_2^2 - 3x_2$ 

 $g_3 = -3x_2^5 - x_2^4 + 2x_2^3 + x_2^2 - 3$ |

Table 8: Success examples ($n = 4$)

| ID | $F$ | $G$ |
|---|---|---|
| 1 | $f_1 = -2x_0x_3^3 - x_3^8 + 2x_3^6 - x_3^5 + x_3^3 + 2x_3$ 
 $f_2 = 3x_0^2x_1x_3^4 - 2x_0x_1x_3^9 - 3x_0x_1x_3^7 - 2x_0x_1x_3^6 + 2x_0x_1x_3^4 - 3x_0x_1x_3^2 + x_0 - 3x_3^5 - x_3^3 + 3$ 
 $f_3 = 2x_0^2x_2^2 + 3x_0x_1^2x_3 + x_0x_2^2x_3^5 - 2x_0x_2^2x_3^3 - x_0x_2^2 - 2x_1^2x_3^6 - 3x_1^2x_3^4 + 2x_1^2x_3 + x_1 + x_3^5 - 3x_3$ 
 $f_4 = -2x_0x_2x_3^2 + x_0x_3^7 + 3x_0x_3^6 - x_0x_3^4$ 
 $f_5 = x_2 + 3x_3^5 + 2x_3^4 - 3x_3^2$ | $g_1 = x_0 - 3x_3^5 - x_3^3 + 3$ 
 $g_2 = x_1 + x_3^5 - 3x_3$ 


 $g_3 = x_2 + 3x_3^5 + 2x_3^4 - 3x_3^2$ 



 $g_4 = -x_3^5 + 2x_3$ |
| 2 | $f_1 = 0$ 
 $f_2 = x_0 - 3x_3^5 - x_3^4 + 1$ 
 $f_3 = -x_0^2x_1x_2 - 2x_0^2x_2x_3^5 + x_0^2x_2x_3^4 + 2x_0^2x_2x_3 - 3x_0^2x_2 - 2x_1x_2^2 + 3x_2^2x_3^5 + 2x_2^2x_3^4 - 3x_2^2x_3 + x_2^2 + x_2 - 2x_3^5 - 2x_3^3 + x_3 + 1$ 
 $f_4 = -3x_0^2x_1^2 + 2x_0x_1^2x_3^5 + 3x_0x_1^2x_3^4 - 3x_0x_1^2 - 2x_0x_3^3 + x_1 - x_3^2x_3^5 + 2x_2^3x_3^4 - 2x_3^3 + 2x_3^5 - x_3^4 - 2x_3 + 3$ 
 $f_5 = -x_0^2x_1^3x_2^2 - 2x_0^2x_1^3x_2 + 3x_0x_1^3x_2^2x_3^5 + x_0x_1^3x_2^2x_3^4 - x_0x_1^3x_2^2 - x_0x_1^3x_2x_3^5 + 2x_0x_1^3x_2x_3^4 - 2x_0x_1^3x_2 - x_0x_1^3 - 3x_0x_1x_2^5 + x_0x_1x_2^4 + 2x_0x_2^2x_3 + 3x_1^3x_3^5 + x_1^3x_3^4 - x_1^3 - 2x_1^2x_2^2 + 3x_1^2x_2 + 2x_1x_2^5x_3^5 + 3x_1x_2^5x_3^4 - 3x_1x_2^5 - 3x_1x_2^4x_3^5 - x_1x_2^4x_3^4 + x_1x_2^4 + 3x_1x_2^2x_3^5 + 2x_1x_2^2x_3^4 - 3x_1x_2^2x_3 + x_1x_2^2 - x_1x_2x_3^5 - 3x_1x_2x_3^4 + x_1x_2x_3 + 2x_1x_2 + x_2^2x_3^6 - 2x_2^2x_3^5 + 2x_2^2x_3 - x_2 + 2x_3^5 + 2x_3^3 - x_3 - 1$ 
 $f_6 = -3x_3^5 - 3x_3^4 + 2x_3^2 - 1$ | $g_1 = x_0 - 3x_3^5 - x_3^4 + 1$ 
 $g_2 = x_1 + 2x_3^5 - x_3^4 - 2x_3 + 3$ 
 $g_3 = x_2 - 2x_3^5 - 2x_3^3 + x_3 + 1$ 


 $g_4 = -3x_3^5 - 3x_3^4 + 2x_3^2 - 1$ |
| 3 | $f_1 = 0$ 
 $f_2 = x_2 + 2x_3^4 + 2x_3 + 1$ 
 $f_3 = x_0 + x_3^5 + 2x_3^4 + x_3 - 3$ 
 $f_4 = -2x_0^2x_1^2 - 3x_0^2 - x_0x_1^2x_2 - 2x_0x_1^2x_3^5 + x_0x_1^2x_3^4 + 3x_0x_1^2x_3 - 2x_0x_1^2 - 3x_0x_1 - x_0x_2^2x_3 - 2x_0x_2x_3^5 - 2x_0x_2x_3^3 - x_0x_2x_3 - 3x_0x_3^5 + x_0x_3^4 - 3x_0x_3 + 2x_0 - 3x_1x_3^5 + x_1x_3^4 - 3x_1x_3 + 2x_1 + 2x_3^4 + 2x_3^2 - x_3$ 
 $f_5 = x_1 + x_3^5 - 2x_3^4 - 3x_3^2 + 2x_3$ | $g_1 = x_0 + x_3^5 + 2x_3^4 + x_3 - 3$ 
 $g_2 = x_1 + x_3^5 - 2x_3^4 - 3x_3^2 + 2x_3$ 
 $g_3 = x_2 + 2x_3^4 + 2x_3 + 1$ 
 $g_4 = 2x_3^4 + 2x_3^2 - x_3$ |
| 5 | $f_1 = x_0 + 3$ 
 $f_2 = -x_0x_2x_3^2 - 3x_2x_3^2 + x_2 - 2x_3^2$ 
 $f_3 = -3x_1x_2^2x_3 - x_1x_2x_3^3 - 2x_3$ 
 $f_4 = 2x_0^3x_1x_2x_3^2 - x_0^2x_1x_2x_3^2 - 2x_0^2x_1x_2 - 3x_0^2x_1x_3^2 - 3x_0x_1^3x_2^2x_3 - x_0x_1^3x_2x_3^3 - 2x_0x_1^2x_3 + x_1 - x_3^4$ | $g_1 = x_0 + 3$ 
 $g_2 = x_1 - x_3^4$ 
 $g_3 = x_2 - 2x_3^2$ 
 $g_4 = -2x_3$ |

Table 9: Success examples ($n = 5$)

| ID | $F$ | $G$ |
|---|---|---|
| 0 | $f_1 = x_0 + 2$
$f_2 = x_3 - x_4^2$
$f_3 = x_1 + 3x_4^2$
$f_4 = x_0^3 x_1 + 2x_0^2 x_1 - 3x_1 x_2^2 - 2x_2^2 x_4^2 - 3x_4^4$
$f_5 = 2x_0^3 x_1^3 x_2 - 3x_0^2 x_1^3 x_2 + x_1^3 x_2^3 + 3x_1^2 x_2^3 x_4^2 + x_1^2 x_2 x_4^4 - 2x_1 x_2 x_3 x_4 + 2x_1 x_2 x_4^3 + x_2$ | $g_1 = x_0 + 2$
$g_2 = x_1 + 3x_4^2$
$g_3 = x_2$
$g_4 = x_3 - x_4^2$
$g_5 = -3x_4^4$ |
| 1 | $f_1 = x_0 - 3x_4^5 - x_4^3 + 3$
$f_2 = 2x_0^2 x_2 + x_0 x_2 x_4^5 - 2x_0 x_2 x_4^3 - x_0 x_2$
$f_3 = -2x_0^2 x_2 x_4 + x_0^2 x_4^6 + 3x_0^2 x_4^5 - x_0^2 x_4^3 - 2x_0 x_1 x_4 - x_1 x_4^6 + 2x_1 x_4^4 + x_1 x_4 - 2x_4^5 + 3$
$f_4 = x_1 + x_4^5 - 3x_4$
$f_5 = x_3 - x_4^5 + 2x_4$
$f_6 = x_2 + 3x_4^5 + 2x_4^4 - 3x_4^2$ | $g_1 = x_0 - 3x_4^5 - x_4^3 + 3$
$g_2 = x_1 + x_4^5 - 3x_4$
$g_3 = x_2 + 3x_4^5 + 2x_4^4 - 3x_4^2$

$g_4 = x_3 - x_4^5 + 2x_4$
$g_5 = -2x_4^5 + 3$ |
| 3 | $f_1 = x_1 x_2^2 x_4 + 2x_1 x_2 x_4^5 + 2x_1 x_2 x_4^2 + x_1 x_2 x_4 - x_2^2 x_3 x_4 - 2x_2 x_3 x_4^5 - 2x_2 x_3 x_4^2 - x_2 x_3 x_4 - 2x_4^5 + 3x_4^3 + x_4 + 1$
$f_2 = x_1 + x_4^5 - 2x_4^4 - 3x_4^2 + 2x_4$
$f_3 = x_3 + 2x_4^4 + 2x_4^2 - x_4$
$f_4 = -3x_1^2 x_2^2 x_3^2 x_4 + x_1^2 x_2 x_3^2 x_4^5 + x_1^2 x_2 x_3^2 x_4^2 - 3x_1^2 x_2 x_3^2 x_4 - 2x_1 x_2^2 x_3^3 x_4 + 3x_1 x_2 x_3^3 x_4^5 + 3x_1 x_2 x_3 x_4^2 - 2x_1 x_2 x_3^3 x_4 - x_1 x_3^2 x_4^5 - 2x_1 x_3^2 x_4^3 - 3x_1 x_3^2 x_4 - 3x_1 x_3^2 - 2x_2^2 x_3^4 x_4 + 3x_2 x_3^4 x_4^5 + 3x_2 x_3^4 x_4^2 - 2x_2 x_3^4 x_4 + x_2 + 3x_3^3 x_4^5 - x_3^3 x_4^3 + 2x_3^3 x_4 + 2x_3^3 + 2x_4^4 + 2x_4 + 1$
$f_5 = -x_0 x_1^2 x_4 - x_0 x_1 x_2 - 3x_0 x_2 x_3 x_4 + x_0 x_2 x_4^5 + x_0 x_2 x_4^3 + 3x_0 x_2 x_4^2 - x_1^2 x_4^6 - 2x_1^2 x_4^5 - x_1^2 x_4^2 + 3x_1^2 x_4 + 2x_1 x_2 x_3 x_4 + 3x_1 x_2 x_4^5 - 2x_1 x_2 x_4^4 - 3x_1 x_2 x_4^3 - 2x_1 x_2 x_4^2 - x_1 x_2 x_4 + 3x_1 x_2 + 3x_4^4 - x_2^3 x_4^4 - x_2^3 x_4 + 3x_3^3 + 2x_2^2 - 3x_2 x_4^4 - 3x_2 x_4 + 2x_2$
$f_6 = x_0 + x_4^5 + 2x_4^4 + x_4 - 3$ | $g_1 = x_0 + x_4^5 + 2x_4^4 + x_4 - 3$


$g_2 = x_1 + x_4^5 - 2x_4^4 - 3x_4^2 + 2x_4$
$g_3 = x_2 + 2x_4^4 + 2x_4 + 1$
$g_4 = x_3 + 2x_4^4 + 2x_4^2 - x_4$










$g_5 = -2x_4^5 + 3x_4^3 + x_4 + 1$ |
| 5 | $f_1 = x_1 - x_4^4$
$f_2 = x_0 + 3$
$f_3 = 2x_0^2 x_1 - 2x_0^2 x_4^4 - x_4^3$
$f_4 = x_3 - 2x_4$
$f_5 = 2x_0^3 x_1^2 x_2 - 2x_0^3 x_1 x_2 x_4^4 - x_0 x_1 x_2 x_4^3 + x_2 - 2x_3 x_4^3 - 3x_4^4 - 2x_4^2$ | $g_1 = x_0 + 3$
$g_2 = x_1 - x_4^4$
$g_3 = x_2 - 2x_4^2$
$g_4 = x_3 - 2x_4$
$g_5 = -x_4^3$ |

Table 10: Failure examples

| ID | $G$ (Ground Truth) | $G'$ (Transformer) |
|---|---|---|
| 1 | $g_1 = x_0 - 3x_1^5 - x_1^3 + 3$ 
 $g_2 = x_1^5 - 3x_1$ | $g_1' = x_0 - 3x_1^5 - x_1^3 + 3$ 
 $g_2' = x_1^5 + 3x_1$ |
| 2 | $g_1 = x_0 - 3x_1^5 - x_1^4 + 1$ 
 $g_2 = 2x_1^5 - x_1^4 - 2x_1 + 3$ | $g_1' = x_0 + 2x_1^5 - x_1^4 + 2$ 
 $g_2' = -x_1^5 - 2x_1^4 - 3x_1 + 2$ |
| 3 | $g_1 = x_0 + x_1^5 + 2x_1^4 + x_1 - 3$ 
 $g_2 = x_1^5 - 2x_1^4 - 3x_1^2 + 2x_1$ | $g_1' = x_0 - x_1^5 + 3x_1^4 + x_1 - 3$ 
 $g_2' = -3x_1^5 - 3x_1^4 - 3x_1^2 + 2x_1$ |
| 4 | $g_1 = x_0 + 3x_1^5 + 3x_1^4 - x_1^3 - 2x_1 - 2$ 
 $g_2 = 3x_1^5 - 2x_1^4 + 2x_1^2 - 1$ | $g_1' = x_0 + 2x_1^5 + 3x_1^4 - 2x_1^3 + 2x_1 - 2$ 
 $g_2' = 3x_1^5 - 2x_1^4 + 2x_1^2 - 1$ |
| 4 | $g_1 = x_0 + 3x_2^5 + 3x_2^4 - x_2^3 - 2x_2 - 2$ 
 $g_2 = x_1 + 3x_2^5 - 2x_2^4 + 2x_2^2 - 1$ 
 $g_3 = -x_2^3 - x_2 - 1$ | $g_1' = x_0 + 3x_2^5 + 3x_2^4 - x_2^3 - 2x_2 - 2$ 
 $g_2' = x_1 + 3x_2^5 - 3x_2^4 - x_2^2 - 1$ 
 $g_3' = -x_2^5 - x_2^4 - x_2^3 - x_2 - 1$ |
| 7 | $g_1 = x_0 + 3x_2^5 + 2x_2^4 + x_2^3 + 3x_2 - 2$ 
 $g_2 = x_1 + x_2^5 + 3x_2^3 + 3x_2^2 + x_2 - 2$ 
 $g_3 = -2x_2^5 - 2x_2^2 + 3x_2 - 1$ | $g_1' = x_0 + 2x_2^5 + 2x_2^4 + x_2^3 + 2x_2^2 + 2x_2$ 
 $g_2' = x_1 + x_2^5 + 3x_2^3 + 3x_2^2 + x_2 - 2$ 
 $g_3' = -2x_2^5 - 2x_2^2 + 3x_2 - 1$ |
| 9 | $g_1 = x_0 - 2x_2^5 + 2x_2^3 + 3x_2^2 - 2x_2 - 2$ 
 $g_2 = x_1 - 2x_2^5 - 3x_2^4 + 2x_2^3 - x_2^2 + x_2$ 
 $g_3 = 3x_2^5 + x_2^3 + 2x_2^2 + 2x_2 + 3$ | $g_1' = x_0 - 2x_2^5 + 2x_2^3 + 3x_2^2 - 2x_2 - 2$ 
 $g_2' = x_1 - 2x_2^5 - 3x_2^4 + 2x_2^3 - x_2^2 + x_2$ 
 $g_3' = -x_2^5 - x_2^3 + 2x_2^2 + 2x_2 + 3$ |
| 15 | $g_1 = x_0 - 3x_2^3 + 2x_2^2$ 
 $g_2 = x_1 - 2x_2^3 - 1$ 
 $g_3 = -x_2^4 + 2x_2$ | $g_1' = x_0 - 3x_2^3 + 2x_2^2$ 
 $g_2' = x_1 - 2x_2^3 - 1$ 
 $g_3' = 2x_2^4 + 2x_2^2$ |
| 4 | $g_1 = x_0 + 3x_3^5 + 3x_3^4 - x_3^3 - 2x_3 - 2$ 
 $g_2 = x_1 + 3x_3^5 - 2x_3^4 + 2x_3^2 - 1$ 
 $g_3 = x_2 - x_3^3 - x_3 - 1$ 
 $g_4 = -3x_3^5 + 3x_3^4 + 2x_3^3 + 2x_3^2 - 3x_3$ | $g_1' = x_0 + 2x_3^5 - 2x_3^4 - 2x_3^3 - 2x_3^2 - 3x_3$ 
 $g_2' = x_1 - x_3^5 - x_3^4 - x_3^2 + 2$ 
 $g_3' = x_2 - x_3^3 - x_3 - 1$ 
 $g_4' = -x_3^5 + 2x_3^4 - 2x_3^3 + 2x_3^2 - 3x_3$ |
| 7 | $g_1 = x_0 + 3x_3^5 + 2x_3^4 + x_3^3 + 3x_3 - 2$ 
 $g_2 = x_1 + x_3^5 + 3x_3^3 + 3x_3^2 + x_3 - 2$ 
 $g_3 = x_2 - 2x_3^5 - 2x_3^2 + 3x_3 - 1$ 
 $g_4 = -x_3^5 - 3x_3^4 + x_3^2 + x_3$ | $g_1' = x_0 + 3x_3^5 + 2x_3^4 + x_3^3 + 3x_3 - 2$ 
 $g_2' = x_1 + x_3^5 + 3x_3^3 + 3x_3^2 + x_3 - 2$ 
 $g_3' = x_2 - 2x_3^5 - 2x_3^2 + 3x_3 - 1$ 
 $g_4' = -2x_3^5 + 3x_3^4 + 3x_3^3 + 3x_3^2$ |
| 9 | $g_1 = x_0 - 2x_3^5 + 2x_3^3 + 3x_3^2 - 2x_3 - 2$ 
 $g_2 = x_1 - 2x_3^5 - 3x_3^4 + 2x_3^3 - x_3^2 + x_3$ 
 $g_3 = x_2 + 3x_3^5 + x_3^3 + 2x_3^2 + 2x_3 + 3$ 
 $g_4 = 3x_3^5 + 2x_3^4 + 3x_3^3 - x_3^2 + 2$ | $g_1' = x_0 - 2x_3^5 + 2x_3^3 + 3x_3^2 - 2x_3 - 2$ 
 $g_2' = x_1 - 2x_3^5 - 3x_3^4 + 2x_3^3 - x_3^2 + x_3$ 
 $g_3' = x_2 + 3x_3^5 + x_3^3 + 2x_3^2 + 2x_3 + 3$ 
 $g_4' = 2x_3^5 + 2x_3^4 + 3x_3^3 + 3x_3^2 + 2x_3$ |
| 10 | $g_1 = x_0 - 3x_3^4 - 3x_3^3 - 2x_3 - 3$ 
 $g_2 = x_1 - x_3^5 - 3x_3^3 - x_3^2 - 3x_3$ 
 $g_3 = x_2 - 3x_3^5 - x_3^4 + 2x_3^3 + x_3^2 - 3$ 
 $g_4 = -3x_3^5 - 2x_3^4 + 2x_3^3 + 2x_3$ | $g_1' = x_0 + 3x_3^4 - 2x_3^3 - 2x_3 - 3$ 
 $g_2' = x_1 - x_3^5 - 3x_3^3 - x_3^2 - 3x_3$ 
 $g_3' = x_2 - 3x_3^5 - x_3^4 + 2x_3^3 + x_3^2 - 3$ 
 $g_4' = -3x_3^5 - 2x_3^4 + 2x_3^3 + 2x_3$ |
| 2 | $g_1 = x_0 - 3x_4^5 - x_4^4 + 1$ 
 $g_2 = x_1 + 2x_4^5 - x_4^4 - 2x_4 + 3$ 
 $g_3 = x_2 - 2x_4^5 - 2x_4^3 + x_4 + 1$ 
 $g_4 = x_3 - 3x_4^5 - 3x_4^4 + 2x_4^2 - 1$ 
 $g_5 = x_4^2 + 3x_4$ | $g_1' = x_0 - 3x_4^5 - x_4^4 + 1$ 
 $g_2' = x_1 + 2x_4^5 - x_4^4 - 2x_4 + 3$ 
 $g_3' = x_2 - 2x_4^5 - 2x_4^3 + x_4 + 1$ 
 $g_4' = x_3 - 3x_4^5 + 2x_4^2 - 1$ 
 $g_5' = x_4^2 + 3x_4$ |
| 4 | $g_1 = x_0 + 3x_4^5 + 3x_4^4 - x_4^3 - 2x_4 - 2$ 
 $g_2 = x_1 + 3x_4^5 - 2x_4^4 + 2x_4^2 - 1$ 
 $g_3 = x_2 - x_4^3 - x_4 - 1$ 
 $g_4 = x_3 - 3x_4^5 + 3x_4^4 + 2x_4^3 + 2x_4^2 - 3x_4$ 
 $g_5 = -x_4^5 - x_4^3 + 2x_4^2 + 2x_4 + 3$ | $g_1' = x_0 + 3x_4^5 + 3x_4^4 - x_4^3 - 2x_4 - 2$ 
 $g_2' = x_1 - x_4^5 - x_4^4 - x_4^2 - x_4 - 1$ 
 $g_3' = x_2 - x_4^5 - x_4^4 - x_4^3 - x_4^2 - 1$ 
 $g_4' = x_3 - 3x_4^5 + 3x_4^4 + 2x_4^3 + 2x_4^2 - 3x_4$ |
| 6 | $g_1 = x_0 + 2x_4^5 - 2x_4^4 + x_4^3 - 2$ 
 $g_2 = x_1 - x_4^5 + 2x_4^3 - 3x_4^2 - 3x_4$ 
 $g_3 = x_2 + 3x_4^5 + 2x_4^4 - 2x_4^3$ 
 $g_4 = x_3 + x_4^5 - 2x_4^3 + 3x_4^2$ 
 $g_5 = 3x_4^5 + 3x_4 + 3$ | $g_1' = x_0 + 2x_4^5 - 2x_4^4 + x_4^3 - 2$ 
 $g_2' = x_1 + 2x_4^5 + 2x_4^3 + 2x_4^2 - 3x_4$ 
 $g_3' = x_2 + 3x_4^5 + 2x_4^4 - 2x_4^3$ 
 $g_4' = x_3 + x_4^5 - 2x_4^3 + 3x_4^2$ 
 $g_5' = 3x_4^5 + 3x_4 + 3$ |

