# OpenReview forum: "Learning to Compute Gröbner Bases"
_ICLR.cc/2024/Conference — Submitted to ICLR 2024_

### Official Review · Reviewer_Sama · 2023-10-18

**Soundness:** 3 good
**Presentation:** 3 good
**Contribution:** 3 good
**Rating:** 8
**Confidence:** 3

**Summary:**

This paper studies the problem of generative model-learned algorithm for Grobner bases. Grobner bases are ubiquitous object for solving polynomial systems, but currently most known libraries that compute Grobner bases do require doubly exponential runtime in the worst case. It is therefore crucial to use tools such as transformers to facilitate this task. This paradigm also poses new interesting problems for Grobner bases: given a Grobner basis, how can one generate a polynomial set that is not a Grobner basis but spanned by the given basis (they call this problem backward Grobner problem)? Moreover, how can one randomly sample a Grobner basis? This paper shows how to solve these problems whenever the polynomials are in shape position and thus for zero-dimensional radical ideals. For the backward Grobner problem, they show a suitable linear mapping can indeed generate a non-Grobner basis while preserving the target set. Their characterization of such linear mapping is rather general, and they show how to sample a subset of these maps via Bruhat decomposition. Finally, experiments are performed, and the accuracy are relatively good. Interestingly, their experiments show that transformers perform much better for lexicographical order instead of graded reverse lexicographical, which is the most popular ordering to my knowledge.

**Strengths:**

This paper studies a very important problem, namely generating Grobner basis for polynomial system solving. Many algorithms for solving polynomial systems and related do require Grobner basis. However, Grobner basis algorithm is notably inefficient, so studying machine learning-based approaches is crucial.

The study of transformer-based approach also elucidates new problems for Grobner bases, such as how to sample them, and how to generate a non-Grobner basis given a Grobner basis. These are novel and intriguing problems and might have further applications. This paper attempts to address the first problem by sampling over zero-dimensional radical ideals, so that most polynomials are in shape position that are easy to handle. It is worth noting that zero-dimensional ideals is a vast and popular family in which many applications lie in. For the backward Grobner problem, they provide a characterization of linear maps that do enable the transformation from Grobner basis to non-basis.

Overall, I think the problems imposed in this paper are interesting, and the theoretical results while not super surprising, are solid.

**Weaknesses:**

While the main selling point is to use transformers for solving the Grobner bases problem, the actual experimental results are not that good. In particular, for the popular grevlex order, the transformer-based approach is very bad. It is surprising that the support accuracy is much higher than the actual accuracy for basis generation. One can argue that the blackbox nature of transformers makes it very hard to interpret the bottleneck of this method, but I do hope other architectures are tried to obtain a better empirical result. This is essential, as the main selling point of this paper is to use machine learning blackbox method to compute Grobner bases more efficiently.

Even though the main motivation of this paper is to use transformers, the theoretical parts and the two problems regarding Grobner bases are intriguing to me than the experiments.

**Questions:**

Typo: on page 2, summary of contributions the second item, it should be "we uncover..." instead of "we uncovered...".

Q: How fast is your transformer-based approach compared to standard algorithms?

---

> ### Author Response · Authors · 2023-11-18
> **To Reviewer Sama**
>
> *(Please kindly check our global response "To all reviewers" beforehand as our reply is based on it.)*
>
> We appreciate your critical reading of our paper and many positive comments on our work.
>
> > ... the actual experimental results are not that good. In particular, for the popular grevlex order, the transformer-based approach is very bad.
>
> Yes, we consider that this demonstrates the challengingness of our newly introduced learning task: addressing NP-hard math problems. The math problems addressed in the prior math transformers are of moderate level. The only exception is a line of work of attacking LWE, the accuracy of which is also far from 100%. The popularity of grevlex in computational algebra derives from its empirical efficiency. Usually, one computes a Gröbner basis with grevlex and then transforms it to lex order using FGLM. Our results suggest that it is better for Transformers to compute lex Gröbner bases directly, which we consider is an interesting observation.
>
>
> > It is surprising that the support accuracy is much higher than the actual accuracy for basis generation.
>
> Support accuracy is a relaxed accuracy metric, so it is not surprising to see that the support accuracy is higher. However, yes, high support accuracy suggests a possibility of hybridization of Transformer and math algorithms, which is one of our next targets.
>
> > One can argue that the blackbox nature of transformers makes it very hard to interpret the bottleneck of this method, but I do hope other architectures are tried to obtain a better empirical result. This is essential, as the main selling point of this paper is to use machine learning blackbox method to compute Grobner bases more efficiently.
>
> Thank you for the important suggestion. We used a vanilla transformer architecture, as in many math transformer studies, so that one can know what and to what extent Transformers can learn by comparing these studies (including ours). As the accuracy is found to be limited, yes, the next step is to design a better architecture. To this end, we have to be careful about the difference in the nature of natural languages and symbolic computations; a correct output for the former can be calculated from the part of the input, while for the latter, the whole input should be taken into account.
>
>
> > Even though the main motivation of this paper is to use transformers, the theoretical parts and the two problems regarding Grobner bases are intriguing to me than the experiments.
>
> We really appreciate this comment, and we do agree with it. As presented in the global response, we consider that one of the core contributions is posing an interdisciplinary challenge from which various studies can appear.
>
>
> **Q1**
> > How fast is your transformer-based approach compared to standard algorithms?
>
> The runtime of Transformer at Table~2 for $n=2, 3, 4, 5$ is 5.879, 11.419, 14.371, 20.622 seconds, respectively. The test batch size was 500, 250, 250, 250, respectively, to fit the input to our 48GB GPU memory. The runtime of Transformer is 5-10 times slower than the math algorithms we tested. Theoretically, the runtime of a Transformer does not depend on the difficulty of the problem as math algorithms do. Shorter encoding of polynomials and better attention mechanisms are important future works. For the former, we may be able to exploit some nature of polynomials; for the latter, sparse attention may not be helpful because, unlike NLP tasks, symbolic computation cares about all the input symbols.

---

> > ### Comment · Reviewer_Sama · 2023-11-18
> >
> > Thank you for your reply! Overall, I think this paper studies a very important problem, and try to solve it using ML models do require much more care than some of the other math problems. This is an interesting perspective, and I appreciate the development of it. Though this paper can be improved in terms of its experiments, I think this paper makes nontrivial contribution to a significantly important math problem. Hence, I raise my score to accept.

---

### Official Review · Reviewer_t6to · 2023-10-31

**Soundness:** 3 good
**Presentation:** 2 fair
**Contribution:** 2 fair
**Rating:** 6
**Confidence:** 2

**Summary:**

This paper discuss using Transformer technology to solve a really hard problem from computational linear algebra: computing the Grobner basis. This problem is known to be NP-hard (with best known double exponential algorithms). Thus, using heuristic methods like ML can be attractive.

Most of the focus of the paper on how to generate the training set. To that end, the authors solve two problems: generating random Grobner bases, and computing non-trivial varities with a prescribed basis. The authors claim that these problems have not be explored before.

I need to qualify my review: I am not an expert, nor even knowledgable, regarding computational algebra. Thus, my review is based on the author's background and claims. I cannot verify the correctness of their claims from an algebraist point of view. This is a low confidence review.

**Strengths:**

- Solving an important problem, which required solving problems not considered before, and allow solving problems that are very expensive to solve without a lot of computational power.
- Part of growing literature on using transformers for symbolic mathematics.
- Good empirical results.
- Well executed study.

**Weaknesses:**

- Limited applicability:
a) The authors acknowledge that transformers may only perform well for in-distribution samples, citing Dziri et al 2023. They do dismiss this as a "a fundamental challenge of transformers, and outside the score, but nevertheless this limits the applicability of their algorithm.
b) In particular, due to the limitations of transformers, they can learn only with instances of bounded size. I am unclear whether this limits also testing, but regardless in means that in-distribution is only of limited size. As a side note, is the problem of finding the Grober basis also NP-hard for bounded size instances?
c) The authors impose additional restrictions on instances in the tranining set, and they claim these are reaslistic.

- Limited theory. although there are some theorems, they are simple and their proofs seem straightforward. The  contribution from the mathematical side seem very limited.

- Algorithm for randomly generating Grobner bases seems very simple, and seem to "engineer" the problem a lot.  There is no analysis of the actual distribution of the instances the algorithm will generate. Is it uniform in any way?

- Contrary to what they authors claim, no real light is shed on the algebraic problems themselves.

- Seems that only experiments on synthetic data was considered.

- Will interest only specialist on work on computational algebra.

**Questions:**

- Did you try you method on "real world" problems?
- I understand that learning can only be done on limited size instances. But does this limit testing as well? If no, why not report experiments.
- On page 6, where you say that you sample O(s^2) polynomails, how are these sampled? What distribution?

---

> ### Author Response · Authors · 2023-11-18
> **To Reviewer t6to (1/2)**
>
> *(Please kindly check our global response "To all reviewers" beforehand as our reply is based on it.)*
>
> We appreciate your critical reading of our paper. We first answer the questions you raised. Replies to other comments will come later or have already been addressed in the global response.
>
> **Q1**
> > Did you try your method on "real world" problems?
>
>  No, because our focus lies on a "general case" rather than special cases that largely vary across applications (see our global response). We believe this is reasonable for the first work. Our work discovered a new challenging machine learning task and new math problems for it, proposed baseline dataset generation method, and provided the experiments using it. We can test the Transformers trained on our dataset to "real world" problems, but such an out-distribution test should result in almost zero accuracy.
>
> **Q2**
> > I understand that learning can only be done on limited size instances. But does this limit testing as well? If not, why not report experiments?
>
> Thank you for your suggestion. Yes, it's limited for now because the maximum sequence length of positional encoding was set based on the maximum sequence length of training samples. We retrained a few models with extended limits for $n=2, 3$. For $n=2$, we generated a new test test for which we set the maximum number of terms of polynomials in $U_1, U_2$ to 4 (previously 2). For $n=3$, we set the density $\sigma=1.0$. We observed a large accuracy drop for such out-distribution samples~(approximately 20-30% decrease). Note that the results relate to out-distribution accuracy, which is beyond the scope of current work (see global response). However, it may be true that the results will serve as a baseline for future work.
>
> **Q3**
> > On page 6, where you say that you sample O(s^2) polynomails, how are these sampled? What distribution?
>
> All the polynomials in this paper are sampled uniformly from the polynomials with a bounded total degree and a bounded number of terms. This is implemented by `random_element` function in SageMath.
>
> **Weakness 1: Limited applicability**
>
> Global response mostly replies to this comment. We admit that our current results are not very practical; however, we'd like to emphasize that this is because the learning of Gröbner basis computation is extremely challenging compared to the math problems addressed in the prior math transformers, requiring many machine learning and mathematical problems to be resolved. Besides, we consider that our task is based more on practical motivation. There is no scalable algorithm for computing Gröbner bases, while there are for the tasks in the related studies. The only exception is the line of works for attacking LWE using Transformers, which is also currently restricted to small problems and not considering generalization.
>
> > a) The authors acknowledge that transformers may only perform well for in-distribution samples, ... nevertheless this limits the applicability of their algorithm.
>
> Yes, and this is actually why prior studies of math transformers explored better training distributions and tested generalization liability. However, this is strongly based on the fact that for their tasks, the random generation of samples is trivial and in-distribution accuracy is already high enough. We found that this is not the case for the Gröbner basis computation and thus focus on this step.
>
> > b) [... The omitted part was answered at Q2. ...] As a side note, is the problem of finding the Grober basis also NP-hard for bounded size instances?
>
> We are not very sure what is meant by "bounded size of instances." An instance size of an NP-hard problem is always bounded. For reference, "computing a Gröbner basis from a given $F$ of finite size" is NP-hard. For example, the polynomials in MQ challenge, a post-cryptography challenge, are degree-2 polynomials. The difficulty of solving degree-2 polynomial systems is the foundation of the multi-variate cryptosystem.
>
> > c) The authors impose additional restrictions on instances in the training set, and they claim these are realistic.
>
> We consider that our setup is reasonable and oriented for practical scenario. Refer to the global response.
>
> **Weakness 2: Limited theory**
> > Limited theory. although there are some theorems, they are simple and their proofs seem straightforward. The contribution from the mathematical side seems very limited.
>
> We have to admit the current theorems and algorithms are simple. This is partially because that the efficiency matters in the large-scale dataset generation. We may design a sophisticated method based on the necessary and sufficient condition (i.e., Theorem 4.7 (3)). Particularly, we may utilize the sygyzy module of $G$, the representatives of which can be efficiently computed as we have an access to a Gröbner basis $G$ (Schreyer's Theorem). We believe that this approach can give a more "uniform" sampling, but it needs a few more steps and is left to our future work.

---

> ### Author Response · Authors · 2023-11-18
> **To Reviewer t6to (2/2)**
>
> **Other Weaknesses**
> > Algorithm for randomly generating Grobner bases seems very simple, and seem to "engineer" the problem a lot. There is no analysis of the actual distribution of the instances the algorithm will generate. Is it uniform in any way?
>
> > Seems that only experiments on synthetic data was considered.
>
> See the global response.
>
> > Contrary to what the authors claim, no real light is shed on the algebraic problems themselves.
>
> Our work introduces a motivation for unexplored algebraic problems and also presents an algorithm and theories for a base case. The intention of this comment is unclear to us, and we would appreciate it if we could have more elaboration.
>
> > Will interest only specialist on work on computational algebra.
>
> We believe that this work may attract interdisciplinary interests. Our new problems should encourage computational algebraists to work on a machine-learning topic. We showcase an example of math problems with difficulty in dataset generation, which should be of interest to the math transformer community. As you've pointed out, our algorithm and theory are designed simply, which may allow readers with limited knowledge of algebra to follow it. The limited in-distribution accuracy at Gröbner basis computation also provides an empirical limit of the Transformer's learnablity, which may interest the researchers in developping a better model. Futher, our work has a potential impact on the cyptography as many cryptosystems are in the scope of *algebraic attack*, which use Gröbner basis computation.

---

### Official Review · Reviewer_DA1f · 2023-11-01

**Soundness:** 4 excellent
**Presentation:** 4 excellent
**Contribution:** 2 fair
**Rating:** 3
**Confidence:** 3

**Summary:**

Grobner basis computation is an important problem in computational algebra, and has applications in cryptography. In this problem we are given as input a non-grobner set, and the output is a grobner basis of the set. The problem is NP-hard, and is also considered to be in hard in practice. This paper investigates the use of transformers in speeding up the solving.

Training a transformer requires a large set of input output pairs from a distribution resembling the distribution of interest,  and this set is not available for the grobner basis problem, due to the computational complexity of generation. To solve this issue, the authors propose a novel method to uniformly sample from the output domain, i.e. set of grobner bases and then find a corresponding input in polytime. The authors then train a transformer on this set and demonstrate the efficacy of their approach. Interestingly, even in the cases where the grobner computation is wrong, the support is correct, which is enough material for other tools to efficiently compute the bases.

**Strengths:**

The paper overall is written and arranged well. The data generation approach proposed by the paper might be of independent interest.

**Weaknesses:**

I found the experimental evaluation unconvincing.

1) The test set seems to be randomly generated instances, rather than problems arising out of some real applications.  Moreover they are generated from the same distribution as the training set. The authors mention the problem of out of distribution generalisation, however it is not addressed at all as far as I could tell.

2) It is usual for solvers to test their performance on some standard datasets (for ex. the SAT competition for SAT solvers), and random instances are usually considered irrelevant. It is not clear whether the state of the art Grobner basis tools have been optimised for random instances or practical instances, hence the comparision may not be fair.

I felt that the use of transformers for this problem is not motivated well enough. The paper does mention that transformers have been used for other math problems, including for a step in the related Buchberger algorithm; however it is not at all clear why they should be used here. Why transformers and not something more basic?

**Questions:**

1) Does the backward generation process induce a uniform distribution on the input space as well?
2) Do typical Grobner basis problems come from use cases which match the distribution of the backward generation?

---

> ### Author Response · Authors · 2023-11-18
> **To Reviewer DA1f (1/2)**
>
> *(Please kindly check our global response "To all reviewers" beforehand as our reply is based on it.)*
>
> We appreciate your critical reading of our paper. Your comments are quite reasonable and have been the focus of the prior studies. However, we would like to emphasize that (we found) learning of Gröbner basis computation has fundamental challenges that other math transformer papers do not have. We have already emphasized it in the global response, so please refer to it. We sincerely ask for your re-evaluation of our work by taking into account our rebuttal.
>
> **Weakness 1 : Out-distribution case**
> > The test set seems to be randomly generated instances rather than problems arising out of some real applications. ... however it [the problem of out-of-distribution generalization] is not addressed ... .
>
> This question is addressed in the global response. In short, the out-distribution case is beyond the scope of this study because, unlike the prior studies, the dataset generation itself is non-trivial, and in-distribution accuracy is not high enough.
>
>
> **Weakness 2: Standard datasets**
> > It is usual for solvers to test their performance on some standard datasets (for ex. the SAT competition for SAT solvers), and random instances are usually considered irrelevant.
>
> This strongly relates to our main claim --- preparing many (non-Gröbner set, Gröbner set) pairs is non-trivial. To our knowledge, there are some classical families of polynomials for benchmarking Gröbner basis computation algorithms (or polynomial system solving algorithms), but these are very far from standard datasets. One may consult [this websight](https://www-sop.inria.fr/coprin/logiciels/ALIAS/Benches/node1.html). The listed examples are artificial, not necessarily zero-dimensional. They are empirically found difficult and/or designed for easily generating variations in the number of variables (e.g., testing for polynomial system Katsura-4 in 4 variables, then Katsura-5 in 5 variables, and so on). Classical benchmarks are useful for math algorithms, i.e., the algorithms proved to work for all the cases (i.e., 100 % "in/out-distribution" samples) except for the timeout, but not for our current work because they are out-distribution samples.
>
> Another benchmark can be found in cryptography challenges (e.g., [MQ Challenge](https://www.mqchallenge.org)). As we have discussed above, the polynomials appearing there are also strongly biased.
>
> > It is not clear whether the state of the art Grobner basis tools have been optimised for random instances or practical instances, hence the comparision may not be fair.
>
> The general purpose algorithms are Buchberger's algorithm and F4 algorithm (SoTA). We tested both of them in our experiments. Thus, the comparison must be fair as the focus of our study is oriented to the general case. Those specialized for particular "practical instances" are not discussed in this paper.
>
> **Weakness 3: Motivation of using Transformer**
>
> In the prior studies, math transformers were studied for testing the learnability of Transformers, rather than using them as practical solvers because we already have well-developed scalable algorithms for the adopted moderate-level math problems.
>
> In contrast, the use of Transformers here is strongly motivated by the fact that there are no scalable mathematical algorithms for computing Gröbner bases. Unlike math algorithms, the runtime of the Transformer does not depend on the difficulty of the problem but on the sequence length. For a large-scale instance on which math algorithms cannot run, Transformers can return an answer with some probability or at least give a hint of the solutions (see discussion on the support accuracy of Table 2). We admit that the current results do not achieve this ultimate goal; resolving this challenge has a long way to go both in machine learning and computational algebra, and this study has initiated an essential step.
>
> > The paper does mention that transformers have been used for other math problems, including for a step in the related Buchberger algorithm;
>
> Quick correction. Transformers have not been used for Gröbner basis computation before. The technique applied to the Buchberger algorithm is simple reinforcement learning. Further, this algorithm assumes binomials and does not work for general polynomials.

---

> > ### Author Response · Authors · 2023-11-18
> > **To Reviewer DA1f (2/2)**
> >
> > **Q1**
> > > Does the backward generation process induce a uniform distribution on the input space as well?
> >
> > We doubt that this is the case, but understanding precisely the distribution of the $F$'s obtained by the backward generation process is a hard mathematical task. To be more precise, $GL_n(k[x_1,\dots,x_n])$, the general linear group over a ring (not a field), is a very complicated group, whose analysis is still open research in the field of group theory (e.g. it is known by a theorem of Suslin that it is generated by elementary matrices, but the amount of factors might be exponential). Hence, understanding orbits upon its action is a hard task. Our study provides a completely new motivation for these mathematical questions.
> >
> > **Q2**
> > > Do typical Grobner basis problems come from use cases which match the distribution of the backward generation?
> >
> > This question has been partially addressed in the global response. Yes, the class of Gröbner bases (more precisely, the class of ideals) tackled by our algorithm is based on the use cases; however, no, the distribution is unknown. Again, this is extremely difficult to know. Gröbner bases are special sets of polynomial sets, the coefficients of which must satisfy some (unknown) algebraic equations. The distribution of Gröbner bases can be roughly and analogically compared to the distribution of matrices with entries constrained by highly nonlinear implicit equations.
> >
> > Both Q1 and Q2 point out that our study raises new mathematical problems that have not been explored previously but are now motivated by machine learning. We will add the discussion on the distribution in the updated version.

---

### Author Response · Authors · 2023-11-18
**Global response (1/2)**

We would like to thank all the reviewers for their time and effort in reading our paper. We carefully read the feedback, many of which are very critical and fruitful, and prepared our rebuttal. We will take into account all the discussion in the update of the manuscript.

Here, we address the comments commonly raised by the reviewers. Note that this reply is very important for all the reviewers, as it aims at reviewers' further understanding (and hopefully, positive re-evaluation) of the contributions of our work.

### **1. Core contributions and novelty**
We would like to emphasize to all the reviewers that our goal of learning Gröbner basis computation is a very challenging and unexplored task. Solving symbolic computation tasks by Transformers has been done by several recent studies. However, we are addressing a task distinguished by the following points.

**Task difficulty.**

Prior studies address the problem of moderate levels such as symbolic integration, ODE solving, polynomial simplification, and linear algebra (to our knowledge, the only exception is a series of works of attacking LWE cryptosystems). This is clearly shown by the facts that (a) scalable mathematical approaches have already been developed and (b) simple learning of Transformers easily achieves almost 100% in-distribution accuracy; this is why the next step, out-distribution accuracy, is of the main focus in prior studies.

While some reviewers are concerned about the motivation and practicality of the learning of Gröbner basis computation, Fact (a) indicates that our task is more practical than the tasks addressed in the prior studies because no scalable Gröbner computational methods exist. Note that we nevertheless believe that studies of learning to solve moderate-level problems are very important as they empirically elucidate the learnability and pitfalls of machine learning models (particularly Transformers). We discuss Fact (b) with the next point.

**Nontriviality in dataset generation.**

To our knowledge, for all the mathematical tasks addressed in the related studies, the dataset generation is more or less trivial (not meaning the generation of a *nice* dataset to generalize to out-distribution is trivial). Further, for most of them, the in-distribution accuracy is almost perfect (Fact (b)). Therefore, the main concern of prior studies lies in increasing out-distribution accuracy. To this end, some studies proposed an improved dataset generation, giving well-tailored sample distribution, or a new Transformer architecture.

In contrast, our study points out for the first time that there exists a problem to which even dataset generation is challenging, and such tasks motivate new mathematical problems to be resolved. Therefore, the central effort of this study has to be put on the stage that is one step behind the other math tasks, which might have appeared "insufficient" to some reviewers with a knowledge of the related studies.

We believe that our work has sufficient motivation and impact, providing an example of a more challenging task for the machine learning (particularly, math transformer) community and encouraging mathematical researchers to independently and/or jointly work on this topic.

---

> ### Author Response · Authors · 2023-11-18
> **Global response (2/2)**
>
> ### **2. Justification of our current setup and real-world applications**
>
> Several reviewers mentioned that our setup appears artificial and may not be based on the practical scenario. This comment is critical, and we'd like to summarize why we reached our current setup.
>
> **Why ideals in shape position (or why zero-dimensional radical ideals)?**
>
> Non-Gröbner sets have various forms across applications. For example, in cryptography (particularly post-quantum cryptosystems), polynomials are restricted to dense degree-2 polynomials and generated by an encryption scheme. On the other hand, in systems biology (particularly, reconstruction of gene regulatory networks), they are typically assumed to be sparse. In statistics (particularly algebraic statistics), they are restricted to binomials, i.e., polynomials with two monomials.
>
> As such, instead of focusing on a particular application, we decided to address the underlying motivations in various applications of computing Gröbner basis: Solving polynomial systems or understanding ideals associated with polynomial systems having solutions.
>
> From this perspective, our restriction to the ideals in shape position is reasonable. Note that it is more mathematically meaningful to focus on the class of ideals than generators (or Gröbner bases). Proposition 4.5 tells us zero-dimensional radical ideals (under a mild condition) are generally in shape position.
>
> Ideals in most practical scenarios are known to be zero-dimensional ideals, and if they are associated with polynomial systems having solutions, they are radical in general (i.e., non-radical ideals are in a zero-measure set in the Zariski topology).
>
> We consider that it is natural to address a general but application-oriented case in the first work. Ideals in shape position (or zero-dimensional radical ideals) are a reasonable choice from this point of view. Specialized methods can come later based on our results.
>
> **On the strategy of sampling ideals.**
>
> It is also worth noting that random sampling of the generators of zero-dimensional ideal requires a specific strategy. In $k[x_1,\ldots,x_n]$ ($k$: infinite field), $s$ degree-bounded polynomials with random coefficients generate (i) (in general) the trivial ideal if $s > n$, the Gröbner basis of which is {1} and (ii) a positive-dimensional ideal with probability 1 if $s < n$. Thus, the only possible way is to consider $s=n$ first and append some combinations of these polynomials to the generators. Our strategy follows this approach. Note that these combinations have to satisfy some algebraic conditions, which is the source of the fundamental difficulty of analyzing the distributions of obtained generators.

---

### Author Response · Authors · 2023-11-21
**Reminder: The discussion period will end soon.**

Dear Chairs and Reviewers

This is a gentle reminder that the discussion period is ending soon. One reviewer responded positively to our rebuttal comments, and two reviewers remain unresponded (or in preparation). Our main rebuttal point is that the math problem and challenges addressed in this study have distinct natures from other math transformer papers. Thankfully, this has been highly evaluated by Review 3 (Review Sama). We hope we can have further feedback from the remaining reviewers; we are willing to resolve any unclarity before the discussion period ends.

We again thank all the chairs and reviewers for their dedication to the review process.

Best regards,
Authors

---

### Meta-Review · Area_Chair_gALX · 2023-12-07

**Metareview:**

The authors initiate the first study to use transformer architecture and an associated ML-based approach in generating Grobner bases. They focus more on the dataset generation problem, which due to the nature of the problem already has several intricacies. The paper's reception was mixed. While many reviewers admitted that the problem is interesting and important, some pointed out the limitations in the study, such as the several strong assumptions (ideal being assumed to be in shape position and zero dimensional) to experimental results not being very representative of the essence of the problem.

This was a very hard call to make. After a lengthy discussion with the SAC, the decision was to not ignore the concerns of Reviewer DA1f. The reviewer was concerned that the random instances used for evaluation may not be representative enough to show the utility of the proposed solution. However, I would like to encourage the authors for a resubmission where they add a basic experiment showcasing utility of the solution by evaluating the trained model on a real problem beyond the synthetically generated random instances.

**Justification For Why Not Higher Score:**

Several assumptions and lack of definitive experimental results to definitely demonstrate the advantage of the proposed transformer-based method.

**Justification For Why Not Lower Score:**

N/A

---

### Decision · Program_Chairs · 2024-01-16

Reject